# Composing Linear Layers from Irreducibles

Travis Pence    Daisuke Yamada    Vikas Singh

University of Wisconsin-Madison
{tnpence, dyamada2}@wisc.edu, vsingh@biostat.wisc.edu

## Abstract

Contemporary large models often exhibit behaviors suggesting the presence of low-level primitives that compose into modules with richer functionality, but these fundamental building blocks remain poorly understood. We investigate this compositional structure in linear layers by asking: *can we identify/synthesize linear transformations from a minimal set of geometric primitives?* Using Clifford algebra, we show that linear layers can be expressed as compositions of bivectors—geometric objects encoding oriented planes—and introduce a differentiable algorithm that decomposes them into products of rotors. This construction uses only $\mathcal{O}\left(\log^2 d\right)$ parameters, versus $\mathcal{O}(d^2)$ required by dense matrices. Applied to the key, query, and value projections in LLM attention layers, rotor-based layers match the performance of strong baselines such as block-Hadamard and low-rank approximations. Our findings provide an algebraic perspective on how these geometric primitives can compose into higher-level functions within deep models.

## 1 Introduction

There is growing consensus [Kozachkov et al., 2023] that, like biological systems, modern models may internally rely on *low-level primitives* that *compose* to form modules with more complex functionality [Weiss et al., 2021]. This compositional perspective was one motivation behind capsule networks [Sabour et al., 2017], which explicitly modeled part-whole relationships through vector-based capsules. But the task of localizing and characterizing such primitives remains challenging albeit interesting. A general capability to compose pre-trained modules in a prescribed way could, in principle, lead to a larger model with predictable and more controllable functionality [Zou et al., 2025, Schug et al., 2024, Ghazi et al., 2019, Abnar et al., 2023, Press et al., 2023], with potential applications ranging from safety guardrails to interpretability.

Some recent results suggest some progress in this direction. In mixture-of-experts architectures [Masoudnia and Ebrahimpour, 2014, Riquelme et al., 2021], specialized sub-networks are conditionally activated and composed via routing [Büchel et al., 2025]. Model merging has evolved from simple parameter averaging to more sophisticated alignment of different model latent representations [Lähner and Moeller, 2024]. Fine-tuning methods like LoRA [Hu et al., 2021] implicitly assume that low-dimensional adjustment of a base network should suffice—essentially a two-level composition. Each approach offers a distinct perspective on composition (e.g., see Chytas et al. [2024]) but are not focused on addressing how the mechanistic composition of low-level primitives gives more complex behavior. To this end, mechanistic interpretability [Rai et al., 2024] and neurosymbolic methods [Yang and Chaudhuri, 2022] explore this space, but are still in a nascent stage of development.

**Scope of this paper.** Consider a core module—with millions of parameters—in a large model. *Can we synthesize its functionality from its most basic primitives? How many such objects would we need?* This casts our broader interest in composition into a concrete problem: *identifying a minimal set of irreducibles that combine in specific ways to realize the full functionality of the module.* We study this problem for *linear layers*—noting that linear layers make up a large portion of parameters in large

39th Conference on Neural Information Processing Systems (NeurIPS 2025).

language models (LLMs). While prior work suggests various parameter-efficient approximations, our interest is not only function approximation, rather to build up the functionality by characterizing its *algebraic structure*. To formalize the above intuition, we use the language of *Clifford algebra*, where linear transformations naturally decompose into simple *bivectors*—geometric objects representing oriented planes. This view reveals how the functionality of a linear layer can be synthesized as a structured composition of a few hundred geometric objects (parameters).

Our key **contributions** are: **(a)** We express linear transformations as compositions of geometric primitives—specifically, bivectors in Clifford algebra—using rotor sandwich products acting on local subspaces of an input multivector. This requires $\mathcal{O}\left(\log^2 d\right)$ scalar parameters, compared to $\mathcal{O}\left(d^2\right)$ for dense layers, where $d$ is the input/output dimension. **(b)** We propose a differentiable invariant decomposition algorithm that maps bivectors to their corresponding rotors in closed-form, which enables integration with `autograd` and gradient-based optimization. **(c)** Empirically, we replace the key, query, and value projections in LLM attention layers and show comparable downstream performance in accuracy and perplexity across various datasets.

Our goal is to show the *feasibility* of this algebraic decomposition approach. It is not a drop-in replacement yet, since practical benefits will require additional system-level integration beyond the scope of this work. The focus here is on the underlying algorithmic and mathematical foundations.

## 2 Preliminaries

We review some relevant concepts from Clifford algebra (also see Hestenes and Sobczyk [2012]).

**Clifford algebra.** A *Clifford algebra*, $\mathrm{Cl}_{p,q}(\mathbb{R})$, is an associative algebra over $\mathbb{R}^n$ equipped with a quadratic form of signature $(p, q)$, where $n = p + q$. The algebra admits two basic products: the *inner product* and *outer (wedge) product*, denoted by $\cdot$ and $\wedge$ respectively. Their sum defines the *geometric product*, which for vectors $u, v \in \mathbb{R}^n$ takes the form:

$$uv \triangleq u \cdot v + u \wedge v.$$

The algebra is generated by orthogonal basis vectors $e_1, \ldots, e_n$, called *generators*, satisfying:

$$e_i^2 = +1 \quad \text{for } 1 \le i \le p; \quad e_j^2 = -1 \quad \text{for } p < j \le n; \quad e_i \wedge e_j = -e_j \wedge e_i \quad \text{for } i \ne j.$$

These relations encode the metric and orientation of the underlying space. The algebra has a canonical basis of $2^n$ elements: the scalar 1 and all distinct products of the basis vectors $e_1, \ldots, e_n$. In particular, *basis bivectors* are wedge products of two distinct basis vectors, i.e., $e_i \wedge e_j = e_i e_j$ for $i < j$. More generally, *basis $k$-vectors* are wedge products of $k$ distinct basis vectors (see Fig. 1), and there are $\binom{n}{k}$ such elements for each $k$. The total number of such basis elements is the *dimension* of the algebra. The *reversion*, given by †, reverses the order of basis vectors. For example, $(e_1 e_2)^\dagger = e_2 e_1$ and $(e_1 e_2 e_3)^\dagger = e_3 e_2 e_1$.

The algebra decomposes into a direct sum of subspaces indexed by *grade*: scalars (grade 0), vectors (grade 1), *bivectors* (grade 2), and general *$k$-vectors*, which are linear combinations of corresponding basis elements. A *multivector* is a general element of $\mathrm{Cl}_{p,q}(\mathbb{R})$, expressed as a linear combination of components of multiple grades. For example, $e_1 e_2$ is a basis bivector; $e_1 e_3 + 2 e_2 e_3$ is a bivector; and $1 + (2e_2 + e_4) - e_1 e_3$ is a multivector composed of elements of grades 0, 1, and 2. We denote the subspaces of vectors and bivectors by $\mathrm{Cl}^1(\mathbb{R})$ and $\mathrm{Cl}^2(\mathbb{R})$, and more generally, $\mathrm{Cl}^k(\mathbb{R})$ for grade-$k$ elements. The algebra naturally splits into *even* and *odd* subalgebras based on grade parity. The *even subalgebra* $\mathrm{Cl}^+(n) \subset \mathrm{Cl}(n)$ consists of elements of even grade (scalars, bivectors, 4-vectors, etc), while odd elements include vectors, trivectors, and so on. Some well-known algebraic systems arise as special cases of Clifford algebras: the real numbers $\mathbb{R} \cong \mathrm{Cl}_{0,0}(\mathbb{R})$, the complex numbers $\mathbb{C} \cong \mathrm{Cl}_{0,1}(\mathbb{R})$, the quaternions $\mathbb{H} \cong \mathrm{Cl}_{0,2}(\mathbb{R})$, and the hyperbolic numbers $\mathrm{Cl}_{1,0}(\mathbb{R})$. This way, Clifford algebras naturally generalize familiar algebraic systems by incorporating geometric structure. To keep notations short, we write $\mathrm{Cl}_{p,q}(\mathbb{R})$ as $\mathrm{Cl}(p, q)$ and only consider $\mathrm{Cl}(n, 0)$, denoted as $\mathrm{Cl}(n)$.

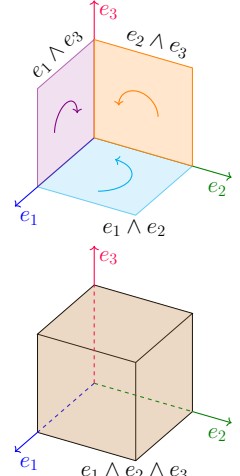

**Figure 1:** The basis vectors, bivectors, and trivector for $\mathrm{Cl}(3)$

# 3 Algebraic Structure of Rotor-based Transformations

We aim to describe standard linear transformations in terms of the algebraic structure of Clifford algebra. We begin by noting how any linear map can be expressed as a sum of multivector products, and then show how restricting to rotors in the Spin group is helpful.

**Clifford form of linear transformations.** A linear map between two vector spaces is one that preserves additivity and homogeneity. Traditionally, such transformations are represented as dense matrices with independent parameters. In contrast, we consider $\mathrm{Cl}(n)$ and write these transformations in terms of the geometric product between multivectors—algebraic objects that encode both magnitude and orientation. We first restate a textbook result.

**Lemma 1.** *(Hestenes and Sobczyk [2012]) Let $a_t$ and $b_t$ denote multivectors in $\mathrm{Cl}(n)$. Any linear function $F$ from $\mathrm{Cl}^k(n)$ to $\mathrm{Cl}(n)$ can be written as the finite sum for some width $w < \infty$,*

$$F(x) = \sum_{t=1}^{w} a_t x b_t. \tag{1}$$

This result shows that linear transformations in Clifford algebra can be expressed as products involving multivectors acting from both the left and right. Thus, Clifford algebra gives us a way to represent a general, arbitrary linear map. But there is a cost: arbitrary multivectors have too much freedom and require all $2^n$ parameters of the full Clifford algebra, which makes the representation inefficient. We will constrain $a_t$ and $b_t$ to preserve *rotational symmetries*, leading to the Spin group

$$\mathrm{Spin}(n) \triangleq \left\{ r \in \mathrm{Cl}^+(n) \mid rr^\dagger = 1 \text{ and } rvr^\dagger \in \mathrm{Cl}^1(n) \text{ for all } v \in \mathrm{Cl}^1(n) \right\}$$

where $\mathrm{Cl}^+(n) \subset \mathrm{Cl}(n)$ is the even subalgebra and $\dagger$ denotes grade-wise reversion. The Spin group captures the set of orientation-preserving rotations within $\mathrm{Cl}(n)$. The elements $r \in \mathrm{Spin}(n)$, called *rotors*, act on multivectors $x \in \mathrm{Cl}(n)$ through the sandwich product as shown in Fig. 2,

$$x \mapsto rxr^\dagger. \tag{2}$$

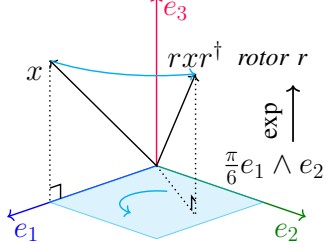

**Figure 2:** The sandwich product rotating a vector $60°$ in the $e_1 \wedge e_2$ plane.

Applying multiple rotors in *parallel* instantiates Lem. 1, where $a_t = r_t$ and $b_t = r_t^\dagger$. We clarify that while the *general* Clifford algebra can represent any linear map, restricting to $\mathrm{Spin}(n)$ limits transformations to orthogonal (rotation-preserving) ones. Consequently, our construction does not capture arbitrary linear maps. In practice, expressivity is recovered by combining multiple rotor modules acting on different subspaces and aggregating their outputs (see Sec. 5). While this sandwich form is useful, we must efficiently parametrize these rotors. To do so, we first observe the relationship between $\mathrm{Spin}(n)$ and the more familiar rotation group, $\mathrm{SO}(n)$.

**Fact 1** (Gallier and Quaintance [2020]). $\mathrm{Spin}(n)$ *is a double cover of the special orthogonal group* $\mathrm{SO}(n)$ *for $n \geq 3$.*

This relationship is important in that while $\mathrm{Spin}(n)$ and $\mathrm{SO}(n)$ are topologically different (hence the "double cover"), they share the same infinitesimal structure—namely, their Lie algebra. This allows using parametrization techniques for $\mathrm{SO}(n)$ to represent elements of $\mathrm{Spin}(n)$. In particular, we will see that rotors can be generated from bivectors via the exponential map, just as rotation matrices arise from skew-symmetric matrices.

**Remark.** *For vector inputs $x$, the sandwich product $rxr^\dagger$ performs the same rotation as the corresponding matrix in $\mathrm{SO}(n)$. However, the full utility of rotors becomes clear when $x$ is multivector: the transformation extends to higher-grade components within the Clifford algebra, going beyond the vector subspace. This makes rotors especially suitable for operating on the richer representations.*

We return to the question: can we efficiently represent rotors so that $a_t$, $b_t$ in Lem. 1 belong to $\mathrm{Spin}(n)$?

**Constructing rotors from bivectors.** *Rotors in $\mathrm{Spin}(n)$ can be parametrized via bivectors*: grade-2 elements of $\mathrm{Cl}(n)$ that encode oriented planes of rotation. To understand this parametrization, we examine the connection between rotors and rotation matrices via Lie groups and their Lie algebras.

**Definition 1.** *A Lie group $G$ is a smooth manifold with the usual group properties along with smooth (infinitely differentiable) group operations. Its associated Lie algebra is a vector space equipped with an antisymmetric, bilinear operation $[X, Y]$, called the Lie bracket, satisfying the Jacobi identity.*

Let us check examples. $\mathrm{Spin}(n)$ and $\mathrm{SO}(n)$ are different Lie groups with the same Lie algebra. Specifically, the Lie algebra of skew-symmetric matrices, $\mathfrak{so}(n) \triangleq \left\{ B \in \mathbb{R}^{n \times n} | B = -B^T \right\}$, underlies both $\mathrm{Spin}(n)$ and $\mathrm{SO}(n)$, despite their topological differences. This shared structure is important. The exponential map gives a surjective correspondence between $\mathfrak{so}(n)$ to $\mathrm{SO}(n)$ as

**Figure 3:** The [bivector $\to$ invariant decomposition $\to$ rotor decomposition $\to$ rotor] process that enables exact parametrization. Note that a *pure* rotor is one that corresponds to a *simple* bivector.

$$\exp(B) = \sum_{i=0}^{\infty} \frac{B^i}{i!}, \qquad (3)$$

showing that every rotation in $\mathrm{SO}(n)$ can be realized as $\exp(B)$ for some $B \in \mathfrak{so}(n)$ using only $\dim(\mathfrak{so}(n)) = \binom{n}{2}$ independent parameters—fewer than the $n^2$ entries of a matrix [Lezcano-Casado and Martínez-Rubio, 2019]. The bridge to the Clifford algebra setting is from another key isomorphism:

**Fact 2** (Doran and Lasenby [2003])**.** *The space of skew-symmetric matrices is isomorphic to that of bivectors, i.e., $\mathfrak{so}(n) \cong \mathrm{Cl}^2(n)$.*

Given a skew-symmetric matrix $B \in \mathfrak{so}(n)$, the corresponding bivector $b \in \mathrm{Cl}^2(n)$ is constructed as

$$b = \sum_{1 \leq i < j \leq n} B_{i,j} \, e_i \wedge e_j.$$

Similar to the exponential map relating $\mathfrak{so}(n)$ and $\mathrm{SO}(n)$ (i.e., $\exp(B)$ generating rotations in $\mathrm{SO}(n)$), every rotor $r \in \mathrm{Spin}(n)$ is the exponential of some bivector $b \in \mathrm{Cl}^2(n)$ given explicitly by the series

$$r = \exp(b) = \sum_{i=0}^{\infty} \frac{b^i}{i!}, \qquad (4)$$

where $b^k$ is the $k$-fold geometric product of $b$ with itself. This means that we can encode a rotor with only $\dim\left(\mathrm{Cl}^2(n)\right) = \binom{n}{2}$ parameters—the dimension of the bivector space.

**Main advantage.** At first glance, our parametrization may seem to offer no advantage. We have parametrized both $\mathrm{Spin}(n)$ and $\mathrm{SO}(n)$ from the same $\binom{n}{2}$ parameters from $\mathfrak{so}(n)$. Notice that while $\mathrm{SO}(n)$ acts only on $n$-dimensional vectors, rotors in $\mathrm{Spin}(n)$ act on a subset of the full $2^n$-dimensional space of multivectors. By utilizing *rotors as the action on multivector input and bivectors as our irreducible primitives*, we will use exponentially fewer parameters.

**Example 3.1.** *A $d \times d$ dense matrix with $d = 2048$, common in self-attention blocks and projection layers (e.g., in LLaMa), uses more than 4M parameters. If $w$ (e.g., $\simeq 3$) is the width hyperparameter in Lem. 1, approximating with bivector irreducibles requires only $w \binom{\log_2 d}{2} = 55 \, w$.*

This reduction (to $55 \times 3$) comes from identifying bivectors as the primitive that generate rich linear maps through composition. Note that the infinite series in (4) is problematic due to the need for approximation. We avoid this by presenting a closed-form solution that preserves differentiability.

## 4 Algorithmic Implementation and Analysis of our Rotor-gadget

We discussed above how rotors can be parametrized through bivectors via the exponential map in (4). A remaining challenge is the infinite series. Of course, we can truncate to a finite length and incur approximation errors. However, for a special class of bivectors, the exponential map admits an exact closed-form solution, which prevents any approximation errors. Moreover, this form remains fully differentiable. We describe the details of this alternative here.

**Algorithm 1** Differentiable Inv. Decomp.

**Require:** $b \in \text{Cl}^2(n)$, $v \in \text{Cl}^1(n)^{k-1}$
**Ensure:** Inv. Decomp. $\{b_1, \dots b_k\}$,
   singular vectors $\{v_1, \dots, v_{k-1}\}$
1: Initialize `decomp`, `vectors` $\leftarrow \emptyset, \emptyset$
2: **for** $i = 1$ to $k - 1$ **do**
3:   $b_s, v_i \leftarrow \text{Proj}_{simple}(b, v_i)$
4:   $b \leftarrow b - b_s$
5:   `decomp` $\leftarrow$ `decomp` $\cup \{b_s\}$
6:   `vectors` $\leftarrow$ `vectors` $\cup \{v_i\}$
7: **end for**
8: `decomp` $\leftarrow$ `decomp` $\cup \{b\}$
9: **return** `decomp`, `vectors`

**Algorithm 2** GA Power Iteration

**Require:** $b \in \text{Cl}^2(n)$, $v \in \text{Cl}^1(n)$,
   threshold $\epsilon \in \mathbb{R}$
**Ensure:** Approximate $\text{Proj}_{simple}(b)$
1: $v_{\text{prev}} \leftarrow v$
2: **while** $\|v + v_{\text{prev}}\| > \epsilon$ **do**
3:   $v_{\text{prev}} \leftarrow v$
4:   $v \leftarrow b \lrcorner (b \lrcorner v)$
5:   $v \leftarrow v / \|v\|_2$
6: **end while**
7: $\sigma u = b \lrcorner v$
8: $b_s \leftarrow \sigma u \wedge v$
9: **return** $b_s, v$

## 4.1 Closed-form differentiable computations of rotors

A key observation is that when a skew-symmetric matrix $B \in \mathfrak{so}(n)$ generates a rotation restricted to a single 2-dimensional plane, the matrix exponential in (3) reduces to a finite closed-form expression—mirroring the simplicity of the classic Rodrigues formula for axis-angle rotations [Goldstein et al., 2002]. An analogous simplification holds for bivectors. The exponential map in (4) admits a closed form when $b \in \text{Cl}^2(n)$ is *simple*, meaning it can be written as $b = u \wedge v$ for some vectors $u, v \in \text{Cl}^1(n)$, or equivalently $b \wedge b = 0$. This ensures that $b$ represents a single-plane rotation rather than a composition of rotations. The resulting closed-form expression is:

$$\exp(b) = \cos(\|b\|) + \frac{\sin(\|b\|)}{\|b\|} b. \tag{5}$$

This form is exact [Doran and Lasenby, 2003]. However, restricting to simple bivectors is limiting, as simple bivectors span only a *subset* of $\text{Cl}^2(n)$, and thus generate only a *subset* of $\text{Spin}(n)$. To capture the full expressivity of rotor-based transformations, we wish to extend to general bivectors.

**Building bivectors from simple bivectors.** To utilize the closed-form exponential in (5) while retaining the full richness of $\text{Cl}^2(n)$, we need a way to express arbitrary bivectors in terms of simple ones. Fortunately, Roelfs and Keninck [2021] show that any bivector $b \in \text{Cl}^2(n)$ admits an *invariant decomposition* as a sum of at most $k \triangleq \lfloor n/2 \rfloor$ mutually commuting, orthogonal, simple bivectors $\{b_1, b_2, \dots, b_k\}$. This decomposition has two key advantages: (1) each component $b_i$ admits an efficient closed-form solution $\exp(b_i)$ in (5) since $b_i$ is simple, and (2) mutual commutativity ensures $\exp\left(\sum_{i=1}^k b_i\right) = \prod_{i=1}^k \exp(b_i)$, via the standard Lie algebra identity $\exp(X + Y) = \exp(X)\exp(Y)$ when their commutator $[X, Y] = 0$. In $\mathfrak{so}(n)$, the Lie bracket is $[X, Y] = XY - YX$, which vanishes when $X$ and $Y$ commute. With these benefits in place, the remaining question is how to construct the decomposition? A recent result provides a spectral formulation in terms of the eigenvectors and eigenvalues of $b$, stated below.

**Lemma 2** (Thm. 4.8 in Eelbode et al. [2024]). *Let $b \in \text{Cl}^2(n)$ have as many eigenvectors as its effective pseudo-dimension. Then, we have*

$$b = \sum_{j=1}^k \mu_j \frac{v_{\mu_j^+} \wedge v_{\mu_j^-}}{v_{\mu_j^+} \cdot v_{\mu_j^-}},$$

*where $\sigma(b) = \{\pm \mu_1, \dots, \mu_k\}$ is the spectrum of $b$ and $v_{\mu_j^+}$ and $v_{\mu_j^-}$ are partner eigenvectors.*

**Differentiable invariant decomposition.** While the invariant decomposition can be computed using eigendecomposition, standard algorithms pose challenges. The eigenvalues of a bivector come in conjugate pairs, and singular values come in positive pairs, making differentiation (and backpropagation) not as straightforward due to the numerical instability of eigen-decomposition for near-degenerate singular values. To address this, we introduce a Krylov subspace-inspired algorithm that iteratively extracts the simple bivectors without requiring explicit eigendecomposition. The

procedure is shown in Alg. 1. It includes a subroutine for projecting a bivector onto the manifold of simple bivectors, which we accomplish with a Clifford algebraic adaptation of the power iteration method shown in Alg. 2. This uses right contraction $b \llcorner v$, which extracts the components of $b$ that lie in the direction of $v$, avoiding the need to construct explicit matrix representations.

The projection has the closed-form expression $\mathrm{Proj}_{\mathrm{simple}}(b) = \sigma(u \wedge v)$ where $\sigma$ is the top singular value of $b$, and $u, v$ are the corresponding left and right singular vectors. Note that because of sign symmetry of the paired singular vectors, we detect convergence by the sum, not their difference, and threshold $\epsilon$. Further discussion and proofs of Alg.1 –2 are in Appendix B. We provide a full visualization of bivector to rotor in Fig. 3.

**Computation graph size.** To control the size of the computation graph, we adapt DEQ [Bai et al., 2019]: we run the fixed-point iteration without gradient tracking, and then perform a single final forward pass with tracking enabled. This ensures that the graph scales only with $k$ (number of components in decomposition), and not the number of iterations in Alg. 2, which depends on the spectral gap and threshold $\epsilon$. Also, while one could initialize $v$ in Alg. 2 using a non-differentiable SVD, this is unnecessary. Under small perturbations of $b$, the singular vectors vary smoothly. This follows from matrix perturbation results, which guarantees that in non-degenerate cases, eigenspaces vary analytically with the matrix entries (Ch. 2 of Kato [1980]). Thus, warm-starting with the previous singular vectors yields fast convergence, if gradient steps are not too large.

## 4.2 A Generalized Rotor Gadget

Occasionally, we may need mappings between arbitrary dimensional spaces. To do so, we now describe a generalized rotor gadget. Instead of the standard sandwich product in (2), we can allow two different rotors on left/right for more expressiveness. Define the rotor-based transformation as:

$$\psi_{r,s}(x) \triangleq r x s^{\dagger}, \qquad (6)$$

where $r, s \in \mathrm{Spin}(n)$. This construction, however, assumes the linear map acts on data with dimension of the Clifford algebra, a power of 2. To address this, we use multiple rotor-sandwich modules operating on different subspaces. For arbitrary input and output dimensions $d_{\mathrm{in}}$ and $d_{\mathrm{out}}$, we utilize the rotor-sandwich modules $\psi_{r,s}(x)$ as building blocks to construct

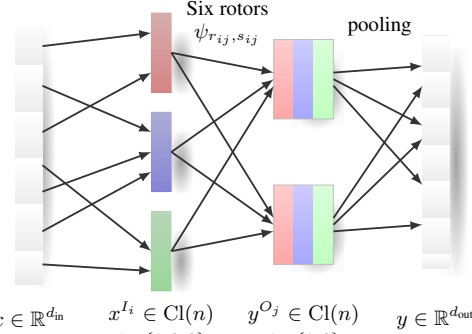

Six rotors $\psi_{r_{ij}, s_{ij}}$    pooling

$x \in \mathbb{R}^{d_{in}}$    $x^{I_i} \in \mathrm{Cl}(n)$    $y^{O_j} \in \mathrm{Cl}(n)$    $y \in \mathbb{R}^{d_{out}}$
$i \in \{1,2,3\}$    $j \in \{1,2\}$

**Figure 4:** Rotor architecture with $c_1 = 3$ and $c_2 = 2$. An input $x$ is split into $\{x^{I_i}\}_{i \in [c_1]}$, each mapped to $y^{O_j}$ via rotor maps $\psi_{r_{ij}, s_{ij}}$, for each $j \in [c_2]$. The outputs $\{y^{O_j}\}$ are pooled and assembled into the final output $y$.

a map $\psi : \mathbb{R}^{d_{\mathrm{in}}} \rightarrow \mathbb{R}^{d_{\mathrm{out}}}$. Fix positive integers $c_1, c_2$, and $n$ with $2^n \leq \min(d_{\mathrm{in}}, d_{\mathrm{out}})$ and let $[h] \triangleq \{1, 2, \dots, h\}$. For each $i \in [c_1]$ and $j \in [c_2]$, define $I_i \subseteq [d_{\mathrm{in}}]$ and $O_j \subseteq [d_{\mathrm{out}}]$ as $2^n$ subsets of input and output coordinates, respectively. We associate each pair with a rotor map $\psi_{r_{ij}, s_{ij}}$ defined by rotors $r_{ij}, s_{ij} \in \mathrm{Spin}(n)$, parametrized by their corresponding bivectors $a_{ij}, b_{ij} \in \mathrm{Cl}^2(n)$. Then, each sub-module operates on $\mathrm{Cl}(n)$ and computes the rotor-sandwich action $\psi_{r_{ij}, s_{ij}} : \mathbb{R}^{I_i} \rightarrow \mathbb{R}^{O_j}$. The full output is defined by aggregating all $c_1 c_2$ rotor maps:

$$\psi(x) \triangleq \sigma\left(\left\{\psi_{r_{ij}, s_{ij}}\left(x^{I_i}\right) \mid i \in [c_1], j \in [c_2]\right\}\right), \qquad (7)$$

where $\sigma$ is a pooling operator on the outputs of $\psi_{r_{ij}, s_{ij}}$. Note that $\cup_i I_i = [d_{\mathrm{in}}]$ and $\cup_j O_j = [d_{\mathrm{out}}]$ are needed to fully cover the input and output dimensions. This gives us a general rotor-based transformation from arbitrary input and output dimensions parametrized by bivectors which is visualized in Fig. 4. We now analyze the number of learnable parameters $\psi$ requires.

**Theorem 1** ($\psi$ Parameter Count). *Let $\psi : \mathbb{R}^{d_{in}} \rightarrow \mathbb{R}^{d_{out}}$ be the mapping defined above, composed of rotor modules $\psi_{r_{ij}, s_{ij}}$ with $i \in [c_1]$ and $j \in [c_2]$, each acting in $\mathrm{Cl}(n)$ with $2^n \leq \min(d_{in}, d_{out}) \triangleq d$. The total number of learnable parameters is upper bounded by*

$$2 c_1 c_2 \binom{n}{2} = \mathcal{O}\left(\log^2 d\right).$$

Thm. 1 shows that rotor maps use only $\mathcal{O}\left(\log^2 \min(d_{\mathrm{in}}, d_{\mathrm{out}})\right)$ parameters. In comparison, a standard dense layer and rank-$r$ factorization require $\mathcal{O}(d_{\mathrm{in}} d_{\mathrm{out}})$ and $\mathcal{O}(r(d_{\mathrm{in}} + d_{\mathrm{out}}))$, respectively.

# 5 Experiments

**Goals.** We empirically evaluate rotors by *replacing key, query, and value linear layers in pre-trained LLMs* and measuring downstream performance on perplexity (PPL) and accuracy. Our experiments span multiple models and datasets. The main goals are to: **(G-1)** Demonstrate the *feasibility of composing linear layers from bivector primitives* by assessing whether rotors match baseline performance of Low-Rank and Block-Hadamard approximations across diverse settings. **(G-2)** Quantify rotor parameter efficiency compared to dense and approximate alternatives. **(G-3)** Analyze how rotor architectural choices—such as width and depth—affect performance.

We focus on linear layers in smaller pre-trained language models (up to 1.5B parameters), where reduced redundancy compared to larger models makes preserving performance harder. We do not attempt full model conversion—which would require larger calibration datasets and a layerwise optimization scheme like GPTQ [Frantar et al., 2022]. Instead, we selectively replace 1-3 attention layers (key, query, and value projections) to isolate the effect, and assess whether rotor decomposition offers competitive PPL and accuracy relative to baselines.

**End-to-end training with rotors.** While the main LLM experiments train rotor layers to *mimic individual dense layers in isolation* (see Sec. 5.1), we also assess their behavior when used throughout a network trained jointly from scratch. To keep the setup lightweight, we replace *all dense layers* in a simple MLP (except the classification head) with rotor layers and train the model *end-to-end* on FMNIST [Xiao et al., 2017] under identical conditions as the dense baseline. Details and results are provided in Sec. 5.2, with experimental settings in Appendix C. With this experiment, we aim to complement the layerwise LLM analysis by testing full-network training with rotor layers.

## 5.1 Experimental Setup

**Substituting attention layers.** Given $x \in \mathbb{R}^d$ to a transformer block, we have:

$$\mathrm{Attn}(x) = \left[\mathrm{softmax}\left(\mathrm{mask}\left(\frac{QK^\top}{\sqrt{d}}\right)\right) V\right] W_o,$$

where the query, key, and value linear projections are defined as $Q = W_q x$, $K = W_k x$, and $V = W_v x$, with $W_q$, $W_k$, and $W_v$ as dense learnable matrices. We *jointly* replace these linear layers in 1–3 selected attention layers with rotors or baseline approximations, *keeping other parameters fixed* except, for consistency, retraining the corresponding output linear layer $W_o$ after each substitution.

**Training protocol and architectural choices.** To fit each substitute layer (rotor, LR, or BH), we extract hidden states from the pre-trained model and minimize MSE between the projected outputs of the original and approximated layers. Each variant is trained independently using the Adam optimizer [Kingma and Ba, 2017]. In our rotor architecture, *depth* refers to the number of stacked rotor maps $\psi$, while *width* denotes the number of parallel rotor maps within each layer. For example, the rotor map in Fig. 4 has both width and depth equal to 1 (i.e., one rotor map). We also insert *fixed permutations* between rotors to enable grade mixing and add *normalization* layers to stabilize training; both are parameter-free. All architectural details and hyperparameters are provided in Appendix C.

|  | Dataset | Method | LLaMa-3.2 1B | | | Qwen-2.5 1.5B | | |
|---|---|---|---|---|---|---|---|---|
|  |  |  | one | two | three | one | two | three |
| Log-PPL | Wikitext2 | Original |  | 2.575 |  |  | 2.287 |  |
|  |  | LR1 | 2.688 | 3.455 | 4.956 | 2.350 | 2.402 | 2.591 |
|  |  | LR4 | 2.658 | 2.729 | 2.880 | 2.342 | 2.372 | 2.548 |
|  |  | BH1 | 2.636 | 2.700 | 2.779 | 2.323 | 2.388 | 2.558 |
|  |  | Rotor | 2.629 | 2.717 | 2.818 | 2.307 | 2.369 | 2.515 |
|  | C4 | Original |  | 3.151 |  |  | 2.834 |  |
|  |  | LR1 | 3.414 | 4.071 | 5.001 | 2.884 | 2.910 | 2.985 |
|  |  | LR4 | 3.390 | 3.315 | 3.504 | 2.874 | 2.905 | 2.980 |
|  |  | BH1 | 3.343 | 3.262 | 3.404 | 2.865 | 2.897 | 2.975 |
|  |  | Rotor | 3.261 | 3.285 | 3.428 | 2.854 | 2.900 | 2.977 |
|  | PTB | Original |  | 3.260 |  |  | 2.985 |  |
|  |  | LR1 | 3.358 | 4.684 | 6.904 | 3.046 | 3.151 | 3.225 |
|  |  | LR4 | 3.316 | 3.400 | 3.466 | 3.034 | 3.127 | 3.192 |
|  |  | BH1 | 3.293 | 3.355 | 3.395 | 3.025 | 3.101 | 3.168 |
|  |  | Rotor | 3.327 | 3.392 | 3.442 | 3.011 | 3.109 | 3.202 |
| Accuracy (%) | Arc Challenge | Original |  | 58.37 |  |  | 66.09 |  |
|  |  | LR1 | 50.78 | 50.44 | 44.26 | 55.06 | 50.97 | 44.55 |
|  |  | LR4 | 53.84 | 53.39 | 45.95 | 57.48 | 54.51 | 60.77 |
|  |  | BH1 | 54.83 | 54.25 | 49.61 | 60.11 | 49.27 | 60.68 |
|  |  | Rotor | 55.31 | 54.50 | 49.64 | 61.34 | 52.27 | 47.28 |
|  | Hellaswag | Original |  | 41.00 |  |  | 55.00 |  |
|  |  | LR1 | 36.17 | 28.93 | 14.47 | 42.53 | 32.33 | 13.93 |
|  |  | LR4 | 38.02 | 33.79 | 33.87 | 44.41 | 40.53 | 11.27 |
|  |  | BH1 | 39.10 | 35.27 | 35.87 | 45.96 | 42.73 | 13.06 |
|  |  | Rotor | 39.33 | 34.94 | 37.52 | 50.20 | 40.60 | 6.868 |

**Table 1:** Log-PPL ($\downarrow$) and accuracy ($\uparrow$) using original, Low-Rank ($r = 1$ or 4), BH1, and Rotor (ours) for 1–3 layer replacements. One-layer results are averaged over all layers; two/three-layer results are averaged over five random selections. Red indicates best, blue second-best per setting.

| Dataset | Method | One layer replaced (Layer index) | | | | | | | | | | | | | | |
|---|---|---|---|---|---|---|---|---|---|---|---|---|---|---|---|---|
| | | **1** | **2** | **3** | **4** | **5** | **6** | **7** | **8** | **9** | **10** | **11** | **12** | **13** | **14** | **15** |
| Wikitext2 (↓) | LR1 | 3.620 | 2.750 | 2.754 | 2.781 | 2.742 | 2.705 | 2.703 | 2.714 | 2.695 | 2.622 | 2.632 | 2.628 | 2.612 | 2.630 | 2.647 |
| | LR4 | 3.851 | 2.723 | 2.752 | 2.673 | 2.673 | 2.650 | 2.671 | 2.667 | 2.674 | 2.620 | 2.628 | 2.614 | 2.602 | 2.620 | 2.634 |
| | BH1 | 3.615 | 2.686 | 2.675 | 2.645 | 2.657 | 2.637 | 2.645 | 2.653 | 2.647 | 2.612 | 2.617 | 2.612 | 2.592 | 2.606 | 2.614 |
| | Rotor | 2.924 | 2.665 | 2.664 | 2.645 | 2.664 | 2.635 | 2.642 | 2.640 | 2.640 | 2.607 | 2.616 | 2.613 | 2.593 | 2.611 | 2.566 |
| C4 (↓) | LR1 | 4.433 | 3.297 | 3.440 | 3.274 | 3.300 | 3.276 | 3.309 | 3.282 | 3.292 | 3.205 | 3.284 | 3.271 | 3.192 | 3.236 | 3.214 |
| | LR4 | 4.432 | 3.292 | 3.404 | 3.250 | 3.266 | 3.250 | 3.264 | 3.260 | 3.281 | 3.203 | 3.204 | 3.191 | 3.187 | 3.219 | 3.208 |
| | BH1 | 4.293 | 3.288 | 3.276 | 3.212 | 3.230 | 3.215 | 3.241 | 3.230 | 3.276 | 3.196 | 3.193 | 3.184 | 3.174 | 3.194 | 3.194 |
| | Rotor | 3.660 | 3.258 | 3.292 | 3.232 | 3.242 | 3.228 | 3.249 | 3.245 | 3.248 | 3.197 | 3.196 | 3.187 | 3.176 | 3.202 | 3.203 |
| PTB (↓) | LR1 | 5.401 | 3.468 | 3.435 | 3.406 | 3.419 | 3.346 | 3.358 | 3.401 | 3.363 | 3.322 | 3.281 | 3.308 | 3.271 | 3.322 | 3.292 |
| | LR4 | 5.183 | 3.394 | 3.395 | 3.316 | 3.347 | 3.304 | 3.315 | 3.334 | 3.328 | 3.281 | 3.276 | 3.265 | 3.264 | 3.308 | 3.287 |
| | BH1 | 4.835 | 3.352 | 3.336 | 3.293 | 3.324 | 3.288 | 3.292 | 3.307 | 3.302 | 3.273 | 3.266 | 3.268 | 3.255 | 3.265 | 3.270 |
| | Rotor | 4.194 | 3.412 | 3.403 | 3.356 | 3.369 | 3.320 | 3.338 | 3.336 | 3.326 | 3.278 | 3.271 | 3.300 | 3.266 | 3.300 | 3.284 |
| Arc Challenge (↑) | LR1 | 50.64 | 53.22 | 46.78 | 46.35 | 45.49 | 51.07 | 52.79 | 51.93 | 33.05 | 50.21 | 56.65 | 55.80 | 56.65 | 55.08 | 55.79 |
| | LR4 | 50.64 | 53.22 | 50.64 | 51.93 | 54.51 | 55.79 | 54.08 | 51.07 | 46.35 | 51.07 | 58.80 | 59.23 | 57.51 | 56.22 | 57.51 |
| | BH1 | 53.22 | 54.51 | 52.79 | 54.94 | 57.08 | 54.08 | 53.65 | 52.36 | 51.93 | 50.21 | 57.94 | 57.94 | 57.08 | 58.80 | 57.51 |
| | Rotor | 54.51 | 55.36 | 53.65 | 55.36 | 54.51 | 55.79 | 54.51 | 53.22 | 52.36 | 50.64 | 58.37 | 60.09 | 56.65 | 57.94 | 57.08 |
| HellaSwag (↑) | LR1 | 29.00 | 32.00 | 34.33 | 39.67 | 34.00 | 33.00 | 34.00 | 33.00 | 34.67 | 37.00 | 41.00 | 38.33 | 40.33 | 40.00 | 39.67 |
| | LR4 | 32.67 | 39.67 | 34.67 | 39.00 | 37.67 | 37.00 | 35.00 | 38.67 | 33.67 | 37.67 | 41.33 | 40.00 | 40.33 | 40.33 | 40.00 |
| | BH1 | 37.33 | 37.67 | 37.67 | 41.00 | 37.00 | 40.00 | 40.33 | 37.00 | 36.33 | 36.33 | 42.67 | 40.00 | 41.33 | 42.00 | 41.33 |
| | Rotor | 40.00 | 39.67 | 35.67 | 42.33 | 41.00 | 38.67 | 36.67 | 38.67 | 36.00 | 36.67 | 42.00 | 39.67 | 40.33 | 42.00 | 40.67 |

| Dataset | Method | Two layers replaced (Layer pairs) | | | | | | | | | | | | | | |
|---|---|---|---|---|---|---|---|---|---|---|---|---|---|---|---|---|
| | | **10,11** | **10,12** | **10,13** | **10,14** | **10,15** | **11,12** | **11,13** | **11,14** | **11,15** | **12,13** | **12,14** | **12,15** | **13,14** | **13,15** | **14,15** |
| Wikitext2 (↓) | LR1 | 2.724 | 2.702 | 2.695 | 2.697 | 2.718 | 2.757 | 2.731 | 2.756 | 2.757 | 2.768 | 2.723 | 2.741 | 2.729 | 2.863 | 2.862 |
| | LR4 | 2.703 | 2.674 | 2.662 | 2.669 | 2.694 | 2.691 | 2.662 | 2.669 | 2.694 | 2.660 | 2.665 | 2.676 | 2.686 | 2.708 | 2.758 |
| | BH1 | 2.677 | 2.660 | 2.645 | 2.650 | 2.656 | 2.670 | 2.643 | 2.662 | 2.660 | 2.645 | 2.656 | 2.657 | 2.655 | 2.680 | 2.700 |
| | Rotor | 2.679 | 2.670 | 2.654 | 2.660 | 2.676 | 2.662 | 2.652 | 2.667 | 2.677 | 2.667 | 2.667 | 2.675 | 2.662 | 2.688 | 2.745 |
| C4 (↓) | LR1 | 3.298 | 3.273 | 3.258 | 3.296 | 3.293 | 3.315 | 3.276 | 3.315 | 3.303 | 3.263 | 3.300 | 3.281 | 3.285 | 3.278 | 3.386 |
| | LR4 | 3.290 | 3.258 | 3.249 | 3.279 | 3.269 | 3.265 | 3.255 | 3.287 | 3.271 | 3.241 | 3.278 | 3.246 | 3.281 | 3.268 | 3.353 |
| | BH1 | 3.281 | 3.241 | 3.228 | 3.246 | 3.244 | 3.242 | 3.227 | 3.246 | 3.244 | 3.216 | 3.237 | 3.226 | 3.234 | 3.232 | 3.296 |
| | Rotor | 3.267 | 3.246 | 3.231 | 3.256 | 3.254 | 3.248 | 3.232 | 3.260 | 3.257 | 3.223 | 3.252 | 3.241 | 3.249 | 3.247 | 3.327 |
| PTB (↓) | LR1 | 3.385 | 3.386 | 3.367 | 3.410 | 3.402 | 3.418 | 3.370 | 3.418 | 3.367 | 3.486 | 3.423 | 3.371 | 3.431 | 3.455 | 3.619 |
| | LR4 | 3.345 | 3.319 | 3.320 | 3.348 | 3.336 | 3.326 | 3.315 | 3.353 | 3.332 | 3.300 | 3.317 | 3.313 | 3.367 | 3.335 | 3.419 |
| | BH1 | 3.315 | 3.307 | 3.297 | 3.304 | 3.310 | 3.309 | 3.287 | 3.332 | 3.301 | 3.281 | 3.293 | 3.298 | 3.294 | 3.300 | 3.345 |
| | Rotor | 3.336 | 3.343 | 3.311 | 3.349 | 3.334 | 3.339 | 3.303 | 3.342 | 3.299 | 3.311 | 3.359 | 3.336 | 3.352 | 3.342 | 3.441 |
| Arc Challenge (↑) | LR1 | 43.06 | 46.07 | 46.50 | 48.21 | 43.92 | 46.93 | 55.51 | 57.23 | 55.94 | 52.94 | 54.65 | 54.65 | 54.22 | 55.51 | 54.65 |
| | LR4 | 42.92 | 48.50 | 48.07 | 51.07 | 49.79 | 59.23 | 56.65 | 59.23 | 57.94 | 58.80 | 56.65 | 57.51 | 56.22 | 55.36 | 57.94 |
| | BH1 | 43.35 | 48.50 | 51.07 | 50.21 | 50.64 | 59.66 | 58.37 | 56.65 | 57.94 | 59.23 | 57.51 | 57.94 | 55.65 | 58.37 | |
| | Rotor | 45.49 | 48.93 | 53.65 | 52.79 | 51.50 | 57.94 | 57.94 | 58.80 | 56.65 | 59.66 | 56.65 | 57.08 | 57.03 | 55.79 | 57.05 |
| HellaSwag (↑) | LR1 | 32.00 | 35.00 | 36.00 | 40.33 | 34.00 | 37.67 | 41.67 | 39.33 | 41.33 | 34.00 | 40.33 | 39.33 | 40.33 | 43.00 | 40.67 |
| | LR4 | 32.33 | 35.33 | 38.33 | 38.33 | 36.67 | 39.00 | 41.67 | 41.00 | 42.33 | 40.00 | 38.00 | 39.00 | 40.00 | 39.00 | 37.67 |
| | BH1 | 37.33 | 37.67 | 35.67 | 40.33 | 39.33 | 39.33 | 41.00 | 41.67 | 41.33 | 40.33 | 43.67 | 40.67 | 42.67 | 40.67 | 40.67 |
| | Rotor | 36.00 | 37.67 | 38.67 | 38.67 | 37.67 | 39.33 | 41.67 | 41.33 | 41.33 | 39.33 | 41.67 | 39.00 | 41.00 | 40.33 | 40.33 |

**Table 2:** Performance on log-PPL (↓) and accuracy (↑) when replacing **one attention layer (top)** for layer indices 1–15 and **two attention layers (bottom)** for pairs of indices from 10–15 of `LLaMa-3.2 1B`. Methods are Low-Rank ($r = 1$ and $4$), BH1, and Rotor.

**Models, datasets, and baselines.** We evaluate on two pre-trained LLMs: `LLaMA-3.2 1B` [Touvron et al., 2023] and `Qwen-2.5 1.5B` [Qwen et al., 2025]. Metrics include log perplexity (↓) on three language modeling datasets—`Wikitext2`, `C4` [Dodge et al., 2021], and `PTB` [Marcus et al., 1993]— and accuracy (↑) on two multiple-choice benchmarks—`Arc Challenge` [Clark et al., 2018] and `HellaSwag` [Zellers et al., 2019]. We compare rotors to: **(a)** *LR1 and LR4*: Low-rank projections with rank $r = 1$ or $4$, where a dense matrix $W \in \mathbb{R}^{d_{\text{out}} \times d_{\text{in}}}$ is approximated as $XY$, with $X \in \mathbb{R}^{d_{\text{out}} \times r}$ and $Y \in \mathbb{R}^{r \times d_{\text{in}}}$. **(b)** *BH1*: Block-Hadamard [Zeng et al., 2023] of depth 1, which approximate $W$ by $BH$, where $H$ is a fixed Hadamard matrix and $B$ is block-diagonal and learnable. Parameter counts for all the methods are detailed in Tab. 4; reference LLM performance (with dense matrices) are shown in Tab. 1. Additional experiments are provided in Appendix D.

## 5.2 Results and Discussion

**Parameter efficiency.** As shown in Tab. 4, rotors require significantly fewer parameters than both dense and approximate baselines. For example, in `LLaMa-3.2 1B`, the query projection uses over 4.19M parameters in its dense form, compared to 16.4K in LR4, 32.7K in BH1, and just $\leq 896$ in rotors—4700× *reduction* over dense and 18× over LR4. Similar savings apply across key and value layers, directly supporting **G-2**.

| #Epoch | 1 | 2 | 3 | 4 | 5 | 6 | 7 | 8 | 9 | 10 |
|---|---|---|---|---|---|---|---|---|---|---|
| Dense | 85.05 | 86.23 | 87.17 | 88.07 | 87.90 | 88.54 | 89.07 | 89.36 | 89.53 | **89.67** |
| Rotor | 80.80 | 82.05 | 82.60 | 82.38 | 86.14 | 86.75 | 86.74 | 86.52 | 87.94 | **88.36** |

**Table 3:** Accuracy (%) on `FMNIST` over 10 epochs for dense and rotor-based MLPs.

**Compute cost.** Rotor layers are currently slower in wall-clock runtime. Depending on rotor width ($w$) and depth ($d$), inference is roughly $w \times d$ times slower than dense layers in our experiments. In terms of FLOPs, rotor layers require $35.8\text{M} \times (wd)$ and $82.4\text{M} \times (wd)$ on `LLaMa-3.2 1B` and `LLaMa-3.2 3B`, compared to 206.1M and 515.3M for dense layers—owing to their block-diagonal structure. These reductions arise naturally from the geometric formulation rather than from explicit sparsity constraints. During training, gradient storage and backpropagation through the decomposition (Fig. 3) add minor additional cost. Optimized kernels and hardware-aware implementations could exploit this structure much better, as well as offer memory-bandwidth benefits discussed in Sec 7.

**Rotor performance vs. baselines.** Despite their compact size, rotors match or outperform LR1, LR4, and BH1 across most datasets and models. In Tab. 1, on `Wikitext2`, rotors achieve 2.629 log-PPL in `LLaMa-3.2 1B` (vs. 2.636 for second-best BH1), and 2.307 in `Qwen-2.5 1.5B` (vs. 2.323 for second-best BH1) when a single attention layer is replaced. On `Arc Challenge`, rotor layers yield 55.31% accuracy in `LLaMa-3.2 1B` (vs. 54.83% for BH1) and 61.34% in `Qwen-2.5 1.5B` (vs. 57.48% for LR4). Rotor projections are consistently either the best or second-best across most settings.

Tab. 2 confirms rotor robustness across layer combinations. For example, replacing layers 12 and 13 in `Qwen-2.5 1.5B` yields 59.66% accuracy on `Arc Challenge`, outperforming LR4 (58.80%) and BH1 (57.94%). In contrast, LR1, despite having 2–5$\times$ more parameters than rotors, shows significant performance drops (e.g., log-PPL on `Wikitext2` for `LLaMa-3.2 1B`). These results confirm **G-1** and show that rotors match downstream performance of other baselines only using far fewer parameters.

In Fig. 5, PPL consistently decreases with both increasing rotor *width* and *depth*. The strongest improvements occur from depth 1 to 2, after which gains taper off. This trend suggests that both stacking more layers (depth) and using more parallel rotor maps (width) contribute to better approximation of linear layers—showing **G-3**.

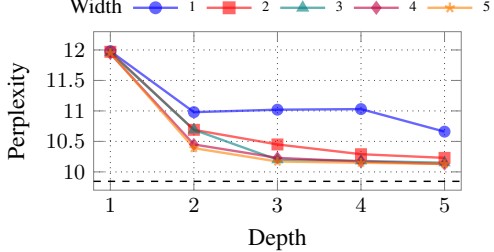

**Figure 5:** Effect of **rotor width and depth**. Replacing Layer-13 in `Qwen-2.5 1.5B` with rotors of varying depth and width. The dashed line (9.845) indicates convergence to the base model's perplexity.

**Results of end-to-end training on `FMNIST`.** As shown in Tab. 3, the dense network improves faster in early epochs, while the rotor-based model learns more gradually but reaches a comparable final accuracy (89.67% vs. 88.36%). This suggests that end-to-end optimization with rotors is slightly slower—potentially due to the high compression—but ultimately competitive with dense layers.

Overall, the LLM results support our claim: rotor layers—constructed from bivector primitives—offer insight into how linear layers can be synthesized from a compact set of building blocks. Moreover, the `FMNIST` experiment shows that rotor layers are feasible when trained jointly with the rest of the network end-to-end from scratch. Our full codebase, including datasets and hyperparameters for all experiments, is available at `https://github.com/vsingh-group/ComposingLinearLayers`. Key architectural and training details are also provided in Appendix C.

## 6 Related Work

**Compositions in machine learning.** Building complex model behavior from composition of simpler compute primitives is an active research topic. Capsule networks [Sabour et al., 2017] modeled part-whole relations, while mixture-of-experts [Riquelme et al., 2021] and model merging [Lähner and Moeller, 2024] compose specialized sub-modules in a conditional manner. Recent studies show how internal circuits in LLMs organize around functional units [Weiss et al., 2021]. In NLP, Tree-structured models [Tai et al., 2015] have been used to encode syntactic composition, while works

on compositional semantics [Mitchell and Lapata, 2008] investigate how word meanings compose to form sentence meanings. The ability to understand new combinations of familiar components remains a challenge [Zhou et al., 2024, Lake and Baroni, 2018] and a few strategies are being studied [Li et al., 2022, Chen et al., 2020, Chytas et al., 2024]. Other directions include modular approaches [Das et al., 2018, Pfeiffer et al., 2023].

**Approximating linear layers.** To reduce the cost of dense layers, various structured approximations are common. Low-rank factorization such as LoRA [Hu et al., 2021] constrains weights to rank-$r$ matrices. Other matrices such as circulant [Yu et al., 2014, Cheng et al., 2015], Toeplitz [Sindhwani et al., 2015], Walsh-Hadamard [Alam et al., 2024], and Block-Hadamard [Zeng et al., 2023], among others, have been shown to be resource efficient. Our goal is to understand the compositional structure of the linear transformation itself. Resource efficiency is the *result*, not the key motivation.

| Method | Key | Query | Value |
|--------|-----|-------|-------|
| Dense | 1048576 | 4194304 | 1048576 |
| LR1 | 2560 | 4096 | 2560 |
| LR4 | 10240 | 16384 | 10240 |
| BH1 | 8192 | 32768 | 8192 |
| Rotor | $\leq 1080$ | $\leq 896$ | $\leq 1080$ |

**Table 4:** Summary of **the number of parameters** used for key, query, and value projections in a single attention layer of `LLaMa-3.2 1B`.

**Clifford/Geometric algebra in ML.** Recent works have explored Clifford algebra for encoding geometric structure in ML models. GCANs [Ruhe et al., 2023b] and CGENNs [Ruhe et al., 2023a] construct equivariant layers by combining GA-based transformations. Clifford neural layers [Brandstetter et al., 2023] was applied to physical modeling tasks. GA has been connected to convex training of ReLU networks [Pilanci, 2024] and randomized methods for multivector representations [Wang et al., 2024] are available. GA has also been used for time-series models [Chen et al., 2025].

## 7 Conclusions and Future Work

We show that the functionality of standard linear layers can be expressed with exponentially fewer parameters, $\mathcal{O}\left(\log^2 d\right)$ versus $\mathcal{O}\left(d^2\right)$, while maintaining competitive performance when applied to attention mechanisms in modern LLMs. In particular, our experiments on 1–1.5B parameter models demonstrate that rotor-based modules can effectively reproduce the behavior of dense linear layers when trained in isolation, while our end-to-end experiment on `FMNIST` shows their feasibility when trained jointly with the rest of the network from scratch. The underlying algebraic structure reveals a rich compositional hierarchy, where geometric primitives combine through rotor operations to form expressive yet compact transformations. This is achieved by mapping bivector parameters to their corresponding rotor operations. These insights open several promising directions, including the development of interpretable architectures and parameter-efficient models that leverage this compositional structure. Beyond that, our framework connects to statistical models involving interaction decomposition, such as ANOVA [St et al., 1989], multivariate analysis [Mardia et al., 2024], and mechanisms to approximate Shapley features [Chen et al., 2023]. For example, modeling high-order dependencies in statistical learning often suffers from exponential parameter growth. While ideas such as Tucker tensor decomposition [Li et al., 2018] factorize the full coefficient tensor, our framework offers an alternative by mapping $k$ predictor variables to a multivector $X$ and modeling the response variable via a rotor transformation as $\hat{y} = \langle \exp(a) X \exp(b)^{\dagger} \rangle_0$ learned from $\mathcal{O}(k^2)$, where $(a, b)$ are learned bivectors from our algorithms. This constructs interaction spaces through the composition of 2-way primitives.

**Practical considerations.** Our current implementation remains computationally demanding, since it does not yet exploit the inherent sparsity or algebraic structure of rotor decompositions (see Appendix A for a discussion of limitations). In parallel, a promising direction is to leverage these ideas to mitigate memory bandwidth bottlenecks that dominate large-model inference. As noted in Davies et al. [2025], (i) memory capacity requirements for frontier LLMs exceed hundreds of gigabytes, and (ii) memory bandwidth, rather than compute, is the primary bottleneck for inference throughput. Our approach directly addresses (ii): instead of loading millions of dense-layer weights from memory, rotor layers can, in principle, synthesize them on-chip from a small set of geometric parameters, effectively trading compute for memory bandwidth. Future work will focus on developing sparse and hardware-aware Clifford kernels to realize these gains in practice and on extending the framework to full-model training and dynamic rotor composition for scalable deployment.

## Acknowledgments

We thank the anonymous reviewers for their constructive feedback. We are very grateful to Prof. Karthikeyan Sankaralingam for numerous valuable discussions and working out the link to Davies et al. [2025]. TP, DY and VS were all partly supported by NIH R01AG092220.

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

## Appendix

In the Appendix, we provide additional discussions, formal proofs, and further experimental details that support the main paper. Section A outlines the limitations of this work. Section B presents the proofs of the algorithms and theorems, along with general results for certain special cases. Section C describes the hyperparameters and training setups used in the experiments. Section D includes additional experimental results.

## A  Limitations

While our results demonstrate feasibility, they remain theoretical at this stage—though rotor layers achieve competitive performance when replacing attention layers, the benefits are primarily in a radical parameter count reduction rather than immediate compute speedup in practice. Realizing such gains will require dedicated systems-level optimizations beyond the scope of this work.

Our current implementation relies on dense matrix representations of rotors and does not yet exploit the inherent sparsity of the rotor decomposition. Leveraging this sparsity will require new backpropagation schemes and software libraries, particularly when training from scratch. We discuss the matrix representation and sparse structure of rotors in detail in Section B.

Overall, this work is intended to highlight the feasibility and promise of decomposing linear layers—key building blocks in modern large models—into smaller (potentially geometric) modules, such as bivectors and rotors, to facilitate more compact and interpretable architectures.

## B  Proofs

### B.1  Correctness Proofs for Main Algorithms

We prove the correctness of Alg. 1 and 2 in the theorems below.

**Theorem 2.** *Given a bivector $b \in \mathrm{Cl}^2(n)$, a random vector $v \in \mathrm{Cl}^1(n)$ that has a non-zero component in the direction of the dominant simple component, and a threshold $\epsilon \in \mathbb{R}$, Alg. 2 returns an approximate simple projection of $b$ in a differentiable way.*

*Proof.* Let $B \in \mathfrak{so}(n)$ be the skew-symmetric matrix corresponding to $b$. We begin by showing that $b \lrcorner v = Bv$ for any vector $v$. Fix an orthonormal basis $\{e_1, \ldots, e_n\}$ of $\mathbb{R}^n$ and let

$$b = \sum_{1 \le i < j \le n} b_{ij}(e_i \wedge e_j) \quad \text{and} \quad v = \sum_{k=1}^{n} v_k e_k,$$

noting that $[B]_{ij} = b_{ij}$ for $1 \le i < j \le n$ and $[B]_{ij} = -b_{ji}$ for $1 \le j < i \le n$. It is easy to verify that $(e_i \wedge e_j) \lrcorner v = v_j e_i - v_i e_j$. By bilinearity of the right contraction, we have

$$
\begin{aligned}
b \lrcorner v &= \sum_{i<j} b_{ij}(v_j e_i - v_i e_j) \\
&= \sum_{i<j} b_{ij}(v_j e_i) - \sum_{i<j} b_{ij}(v_i e_j) \\
&= \sum_{i<j} b_{ij}(v_j e_i) - \sum_{j<i} b_{ji}(v_j e_i) && \text{swapping } i \text{ and } j \\
&= \sum_{i<j} [B]_{ij}(v_j e_i) + \sum_{j<i} [B]_{ij}(v_j e_i) \\
&= \sum_i \sum_j [B]_{ij} v_j \, e_i = \sum_i (Bv)_i e_i = Bv.
\end{aligned}
$$

It follows that

$$b \lrcorner (b \lrcorner v) = b \lrcorner Bv = B^2 v = -B^T B v.$$

Since $B^2$ is symmetric, we may apply the power iteration method. Observe, by skew-symmetry, the non-zero eigenvalues of $B$ come in conjugate pairs

$$\{\pm i\sigma_1, \pm i\sigma_2, \ldots, \pm i\sigma_k\},$$

with $\sigma_1 \geq \sigma_2 \geq \cdots \geq \sigma_k$ and $k \leq \lfloor n/2 \rfloor$. An associated orthonormal set of complex eigenvectors can be written as $\{u_j \pm iv_j\}_{j \leq k}$. Since $\sigma_1$, the dominant eigenvalue for $B^2$, has multiplicity 2, the power method on $B^2$ will not converge in a single direction. Instead, it will *oscillate* within the two-dimensional span of $\{u_1, v_1\}$ Wilkinson [1965].

Let $v$ be one of the singular vectors on which the algorithm terminates. Since singular vectors are unique up to rotation, WLOG we choose $v$ to be the right singular vector. Then, the left singular vector $u$ and singular value $\sigma$ satisfy $b \lrcorner v = Bv = \sigma u$ by definition.

By the Eckart-Young-Mirsky theorem [Golub and Van Loan, 2013], the best rank-2 approximation to $B$ is its projection onto its simple two-dimensional subspace spanned by top two singular directions:

$$\text{Proj}_{\text{simple}}(b) = \sigma_1 u_1 v_1^T + \sigma_2 u_2 v_2^T.$$

As the singular values come in pairs for $B$, this reduces to $\sigma_1 \left( u_1 v_1^T + u_2 v_2^T \right)$. But for $B$, the two right singular vectors $v_1, v_2$ and the two left singular vectors $u_1, u_2$ lie in the same two-plane, coming from the dominant eigenvalue conjugate pair, $\pm i\sigma_1$. Thus, we can choose $u_2 = v_1$ and $v_2 = -u_1$ (or vice versa), which ensures $u_1 \perp v_1$ and $u_2 \perp v_2$. It follows that

$$\text{Proj}_{\text{simple}}(b) = \sigma_1 \left( u_1 v_1^T - v_1 u_1^T \right) = \sigma_1 (u_1 \wedge v_1).$$

Since Alg 2 solves only approximately for $\sigma(u \wedge v) \approx \sigma_1(u_1 \wedge v_1)$, we have that

$$\text{Proj}_{\text{simple}}(b) \approx \sigma_1(u_1 \wedge v_1).$$

Note that since the right contraction implementation is differentiable, Alg 2 is differentiable. $\qquad\square$

**Theorem 3.** *Given $b \in \text{Cl}^2(n)$ and $k - 1$ many vectors $v \in \text{Cl}^1(n)$, Alg. 1 returns the invariant decomposition in a differentiable way.*

*Proof.* The heavy lifting is done by Eelbode et al. [2024] in Lem. 2, which states that $b$ can be written as the sum of at most $k$ commuting, orthogonal, simple bivectors

$$b = \sum_{j=1}^{k} \mu_j \frac{v_{\mu_j^+} \wedge v_{\mu_j^-}}{v_{\mu_j^+} \cdot v_{\mu_j^-}}$$

where $\{\pm\mu_1, \pm\mu_2, \ldots, \pm\mu_k\}$ is the spectrum of $b$ and $v_{\mu_j^+}$ and $v_{\mu_j^-}$ are partner eigenvectors.

To obtain the full decomposition, it suffices to iteratively extract each term. If we have a differentiable subroutine to find the first term, then we can subtract it from $b$, apply the same routine to the residual, and repeat. Thus, it suffices to prove that the first term in the sum is equal to the projection of $b$ onto the simple bivectors.

From the proof of Alg. 2, we know that

$$\text{Proj}_{\text{simple}}(b) = \sigma(u \wedge v)$$

where $\sigma$ is the largest singular value and $u, v$ are the corresponding left and right singular vectors. We wish to show that

$$\sigma(u \wedge v) = \mu_1 \frac{v_{\mu_1^+} \wedge v_{\mu_1^-}}{v_{\mu_1^+} \cdot v_{\mu_1^-}}.$$

The partner eigenvectors take the form $v_{\mu_1^+} = u + iv$ and $v_{\mu_1^-} = u - iv$, as cited in the proof of Alg. 2. Then, the numerator reduces to

$$
\begin{aligned}
v_{\mu_1^+} \wedge v_{\mu_1^-} &= (u + iv) \wedge (u - iv) \\
&= u \wedge u - iu \wedge v + iv \wedge u + v \wedge v \\
&= -iu \wedge v + iv \wedge u \\
&= -2iu \wedge v.
\end{aligned}
$$

as the wedge product is antisymmetric. The denominator reduces to

$$
\begin{aligned}
v_{\mu_1^+} \cdot v_{\mu_1^-} &= (u + iv) \cdot (u - iv) \\
&= u \cdot u - iu \cdot v + iv \cdot u + v \cdot v \\
&= u \cdot u + v \cdot v \\
&= 2
\end{aligned}
$$

as singular vectors are orthonormal. Since $\mu_1 = i\sigma$, we have that

$$\mu_1 \frac{v_{\mu_1^+} \wedge v_{\mu_1^-}}{v_{\mu_1^+} \cdot v_{\mu_1^-}} = i\sigma \frac{-2iu \wedge v}{2} = \sigma(u \wedge v) = \mathrm{Proj}_{\mathrm{simple}}(b),$$

as desired. Subtraction is differentiable and as $\mathrm{Proj}_{\mathrm{simple}}(b)$ can be found in a differentiable way, this algorithm is differentiable. $\qquad\square$

## B.2 Parameter Count

We revisit the main theorem on the parameter count for rotor maps and provide a proof of the statement.

**Theorem 4** ($\psi$ Parameter Count). *Let $\psi : \mathbb{R}^{d_{in}} \to \mathbb{R}^{d_{out}}$ be a linear map composed of rotor modules $\psi_{r_{ij},s_{ij}}$ with $i \in [c_1]$ and $j \in [c_2]$, each acting in $\mathrm{Cl}(n)$. Let $2^n \le d \triangleq \min(d_{in}, d_{out})$. Then, the total number of learnable parameters is upper bounded by*

$$2c_1 c_2 \binom{n}{2} = \mathcal{O}(\log^2 d).$$

*Proof.* Each rotor is the exponential of a bivector. As general bivectors are a linear combination of basis bivectors, of which there are $\binom{n}{2}$, parametrizing a bivector takes $\binom{n}{2}$ scalar parameters. Hence, a single rotor-sandwich map $\psi_{r_{ij},s_{ij}}$ is parametrized using $2\binom{n}{2}$ scalar parameters. The result is then immediate, since there are $c_1 c_2$ such modules, corresponding to all input and output pairs. $\qquad\square$

Tab. 7 summarizes the number of parameters required for each key, query, and value projection per attention layer using different replacement methods across all LLMs used in our experiments.

## B.3 Technical Analysis of Matrix Representation of Rotor Application

We implement the sandwich product in (2), along with other operations such as grade-restricted wedge and inner products, using the `torch_ga` Clifford algebra package for PyTorch, available at Alesiani. In our implementation of (2), the rotor action is represented by a matrix $M$. In this section, we describe the construction of $M$ and introduce some of its key properties. These results highlight the special orthogonality and block structure of $M$. While the two-rotor map $\psi_{r,s}(x)$ in (6) does not satisfy these properties exactly, it appears to share similar properties and structure; we do not discuss it here.

We compute the sandwich product using a change-of-basis matrix.

**Definition 2.** *Let $r \in \mathrm{Spin}(n)$, and let $\{e_J\}_{J \subseteq [n]}$ denote the canonical basis of $\mathrm{Cl}(n)$ (where $J$ is an ordered multi-index). Define*

$$N_r = [\tau(\psi_r(e_J))]_{J \subseteq [n]},$$

*where $\psi_r(x) = rxr^\dagger$ and $\tau$ is the canonical vector-space isomorphism $\tau : \mathrm{Cl}(n) \to \mathbb{R}^{2^n}$. That is, each row of $N_r$ is the coefficients of $re_J r^\dagger$ expressed in the basis. We refer to $N_r$ as the change-of-basis matrix for $r$.*

**Lemma 3.** *Let $x \in \mathrm{Cl}(n)$, $r \in \mathrm{Spin}(n)$, and $N_r$ be defined as in Def. 2. Then, the two mappings*

$$x \mapsto x N_r \quad and \quad x \mapsto rxr^\dagger$$

*are the same linear map up to isomorphism.*

*Proof.* Write $x$ as

$$x = \sum_{J \subseteq [n]} x_J e_J,$$

where $x_J$ is the real coefficient of the basis element $e_J$. Let $\tau : \mathrm{Cl}(n) \to \mathbb{R}^{2^n}$ be the canonical vector-space isomorphism that gives the row vector coordinates of the multivector, $\phi_r(x) = rxr^\dagger$ for $x \in \mathrm{Cl}(n)$, and $\delta(y) = y N_r$ for $y \in \mathbb{R}^{2^n}$. To prove $\phi$ and $\delta$ are the same up to isomorphism, we must prove that $\tau(\phi_r(x)) = \delta(\tau(x))$ for all $x \in \mathrm{Cl}(n)$.

For each basis element $e_J$, the sandwich product gives some multivector $\psi(e_J) = re_Jr^\dagger$, which we express as

$$re_Jr^\dagger = \sum_{I \subseteq [n]} [N_r]_{J,I} e_I,$$

since $(N_r)_{J,I}$ stores the coefficient of $e_I$ in $re_Jr^\dagger$ by definition. Then,

$$
\begin{aligned}
\tau(\phi_r(x)) &= \tau\left(rxr^\dagger\right) \\
&= \tau\left(r\left[\sum_{J \subseteq [n]} x_J e_J\right] r^\dagger\right) \\
&= \tau\left(\sum_{J \subseteq [n]} x_J re_Jr^\dagger\right) \\
&= \tau\left(\sum_{J \subseteq [n]} x_J \left[\sum_{I \subseteq [n]} [N_r]_{J,I} e_I\right]\right) \\
&= \tau\left(\sum_{I \subseteq [n]} \left[\sum_{J \subseteq [n]} x_J [N_r]_{J,I}\right] e_I\right) \\
&= \sum_{I \subseteq [n]} \left[\sum_{J \subseteq [n]} x_J [N_r]_{J,I}\right] \tau(e_I) \quad \text{by linearity} \\
&= \tau(x) N_r \\
&= \delta(\tau(x))
\end{aligned}
$$

where the second to last step follows as the inner sum is the dot product of $\tau(x)$ with the $I$th column of $N_r$. $\qquad\square$

This allows us to compute the rotor sandwich product via matrix multiplication. However, this gives us very little insight into the structure of $N_r$ itself. For one, $N_r$ must respect the grade-preserving property of rotors, meaning it is a block diagonal matrix. The following alternative view provides more insight into its structure.

**A more revealing view of $N_r$.** Let $b \in \mathrm{Cl}^2(n)$ be the bivector associated with $r \in \mathrm{Spin}(n)$ by $r = \exp(b)$, and let $B \in \mathfrak{so}(n)$ be the corresponding skew-symmetric matrix. Let $R \triangleq \exp(2B) \in SO(n)$ via the matrix exponential map. For each grade $0 \leq k \leq n$, set

$$M_k \triangleq C_k(R) \in \mathbb{R}^{\binom{n}{k} \times \binom{n}{k}},$$

where $C_k(R)$ is the $k$th compound matrix of $R$. Algebraically, $C_k(R)$ is the $k$th exterior power of $R$, i.e., the unique linear map $\bigwedge^k(R)$ such that

$$\bigwedge^k(R)(m_1 \wedge m_2 \wedge \cdots \wedge m_k) = Rm_1 \wedge Rm_2 \wedge \cdots \wedge Rm_k$$

for all $m_i \in \mathbb{R}^n$ for $i \in [k]$ [Conrad, n.d.].

**Definition 3.** *Let $M_k$ be the $k$th compound, or exterior power, of $R$. Stack the $M_k$ on the diagonal to form*

$$M \triangleq \mathrm{diag}(M_0, M_1, \ldots, M_n) \in \mathbb{R}^{2^n \times 2^n}.$$

We first show that $N_r$ and $M$ are in fact the same matrix.

**Theorem 5.** *Let $N_r$ and $M$ be defined as in Def. 2 and 3. Then, $M = N_r$.*

*Proof.* The rotor sandwich product is a grade-preserving automorphism, implying $\psi_r(\mathrm{Cl}^k(n)) = \mathrm{Cl}^k(n)$. Thus, $N_r$ is block-diagonal, one block per grade. First, we show $\psi_r(v) = Rv$ for a vector $v$, where $R = \exp(2B)$. Observe

$$rvr^\dagger = e^b v \left(e^b\right)^\dagger = e^b v e^{b^\dagger} = e^b v e^{-b},$$

where the second equality follows as reversion is an anti-automorphism and the third equality follows as $b^\dagger = -b$. Now, consider the adjoint operator in an associative algebra defined as $\mathrm{ad}_X(Y) \triangleq [X, Y] = XY - YX$. The Hadamard lemma and Taylor expansion together imply

$$e^b v e^{-b} = \sum_{k \geq 0} \frac{1}{k!} (\mathrm{ad}_b)^k(v) = \exp(\mathrm{ad}_b)v.$$

We also note the following identity:

$$\mathrm{ad}_b(v) = bv - vb = (b \cdot v + b \wedge v) - (v \cdot b + v \wedge b) = 2b \cdot v = 2Bv,$$

where $B$ is the skew-symmetric matrix for $b$. Putting these together, it follows that $\psi_r(v) \triangleq rvr^\dagger = \exp(2B)v = Rv$. Now, consider any $k$-vector $v = e_{i_1} \wedge \cdots \wedge e_{i_k}$. We have

$$\begin{aligned} \psi_r(v) &= \psi(e_{i_1}) \wedge \cdots \wedge \psi(e_{i_k}) && \psi_r \text{ is a grade-preserving automorphism} \\ &= (Re_{i_1}) \wedge \cdots \wedge (Re_{i_k}) && \text{from above} \\ &= \overset{k}{\bigwedge}(R)(v). \end{aligned}$$

Since the mapping is unique, we are done. $\qquad\square$

Note that this block-diagonal structure highlights the *grade-preserving behavior of rotors*. This motivates our design choice to allow grade-mixing in our gadget for more expressivity. In particular, we insert random permutations between rotor layers so that information can mix across different grade components of the input multivector. Now that we have an alternative characterization of the sandwich product, we analyze its structure and the space in which it lives. Below, we examine several structural properties of the matrix $M$.

**Property 1.** *The matrix $M$ in Def. 3 is a block diagonal matrix with at most $\binom{2n}{n}$ non-zero entries.*

*Proof.* $M = N_r$ by Thm. 5 and so $M$ is block diagonal. There are $n + 1$ blocks, each of size $\binom{n}{k} \times \binom{n}{k}$ for $0 \leq k \leq n$. Thus, the total number of non-zero entries is at most

$$\sum_{k=0}^n \binom{n}{k}^2 = \binom{2n}{n},$$

which is far less than the $2^{2n}$ of a dense $2^n \times 2^n$ matrix. $\qquad\square$

**Property 2.** *For every grade $0 \leq k \leq n$, the block $M_k = C_k(R)$ lies in $\mathrm{SO}\left(\binom{n}{k}\right)$. Consequently, $M \in \mathrm{SO}(2^n)$.*

*Proof.* By assumption, $B \in \mathfrak{so}(n)$, and so $\exp(2B) = R \in \mathrm{SO}(n)$. Taking the $k$th exterior power preserves orthogonality and a determinant of 1. For each $k$, the $k$th compound matrix $M_k = C_k(R)$ is the matrix representation of the $k$th exterior power $\bigwedge^k(R)$ with respect to the standard basis of $k$-vectors. Since $R$ is orthogonal, $\bigwedge^k(R)$ preserves the induced inner product on $k$-vectors, making $M_k$ orthogonal. Moreover, since $\det(R) = 1$, we have $\det(M_k) = \det\left(\bigwedge^k(R)\right) = (\det(R))^{\binom{n-1}{k-1}} = 1$. Therefore, $M_k \in SO\left(\binom{n}{k}\right)$ for all $0 \leq k \leq n$. To prove $M \in \mathrm{SO}(2^n)$, we note that since $M$ is block diagonal, $M^T = \mathrm{diag}\left(M_0^T, \ldots, M_n^T\right)$ and so $M^T M = I$ as the $M_k$ are special orthogonal. $\det(M) = 1$ as the determinant of block diagonal matrices is the product of the determinant of the blocks, meaning $\det(M) = \prod_{k=0}^n \det(M_k) = \prod_{k=0}^n 1 = 1$. $\qquad\square$

Note that under this view, we obtain a map from $\mathfrak{so}(n)$ to $\mathrm{SO}(2^n)$ by first applying the surjective exponential map from $\mathfrak{so}(n)$ to $\mathrm{SO}(n)$, and then lifting $\mathrm{SO}(n)$ to $\mathrm{SO}(2^n)$ via the exterior powers. This transformation enables an *exponential reduction in the number of parameters required to represent rotor layers*. We conclude by identifying the class of matrices which $M$ belongs. The following is a simple but useful characterization. The group of invertible elements of $\mathrm{Cl}(n)$ is

$$\mathrm{Cl}^\times(n) = \left\{ a \in \mathrm{Cl}(n) | \exists a^{-1} \in \mathrm{Cl}(n) \text{ st. } aa^{-1} = a^{-1}a = 1 \right\}.$$

Let

$$\phi : \mathrm{Cl}^\times(n) \to \mathrm{GL}(2^n, \mathbb{R})$$

be the mapping which assigns to each $a \in \mathrm{Cl}^\times(n)$ its change-of-basis matrix $N_a$. Then for any $N_r$ corresponding to $r \in \mathrm{Spin}(n)$,

$$N_r \in \phi(\mathrm{Spin}(n)).$$

We now give a more refined version of Prop. 2.

**Property 3.** *Let $\phi$ be defined as above. Then,*

$$\phi(\mathrm{Spin}(n)) = \mathrm{SO}\left(2^n\right) \cap \phi\left(\mathrm{Cl}^\times(n)\right) \cap \left( \bigoplus_{k=0}^{n} \mathbb{R}^{\binom{n}{k} \times \binom{n}{k}} \right).$$

*Proof.* The first direction, $\phi(\mathrm{Spin}(n)) \subseteq \mathrm{SO}(2^n) \cap \phi\left(\mathrm{Cl}^\times(n)\right) \cap \left( \bigoplus_{k=0}^{n} \mathbb{R}^{\binom{n}{k} \times \binom{n}{k}} \right)$, is immediate. If $r \in \mathrm{Spin}(n)$, then by definition $r$ is an invertible element in $\mathrm{Cl}^+(n)$ which satisfies $rr^\dagger = 1_s$ (where $1_s$ denotes the scalar identity). Thus, $\mathrm{Spin}(n) \subseteq \mathrm{Cl}^\times(n)$, which implies $\phi(\mathrm{Spin}(n)) \subseteq \phi\left(\mathrm{Cl}^\times(n)\right)$. Furthermore, Prop. 2 and Thm 5 state $N_r \in \mathrm{SO}(2^n)$ for $r \in \mathrm{Spin}(n)$, which ensures $\phi(\mathrm{Spin}(n)) \subseteq \mathrm{SO}(2^n)$. As rotors are grade preserving, $\phi(\mathrm{Spin}(n)) \subseteq \left( \bigoplus_{k=0}^{n} \mathbb{R}^{\binom{n}{k} \times \binom{n}{k}} \right)$.

For the other direction $\mathrm{SO}(2^n) \cap \phi\left(\mathrm{Cl}^\times(n)\right) \cap \left( \bigoplus_{k=0}^{n} \mathbb{R}^{\binom{n}{k} \times \binom{n}{k}} \right) \subseteq \phi(\mathrm{Spin}(n))$, we start by taking an arbitrary element from the set on the LHS and show it must also be in the set on the RHS. Thus, we assume $g \in \mathrm{Cl}^\times(n)$ (an invertible multivector) such that its corresponding matrix $N_g = \phi(g)$ is in $\mathrm{SO}(2^n)$. We wish to show $g \in \mathrm{Spin}(n)$. To do this, it suffices to prove three conditions: $gg^\dagger = 1_s$ (i.e., $g$ is unit-norm), $g \in \mathrm{Cl}^+(n)$ (i.e., $g$ is an even multivector), and $g$ is grade preserving.

The matrix $N_g$ represents the linear transformation $T_g(X) = gXg^\dagger$. The condition $N_g \in \mathrm{SO}(2^n)$ means that $N_g$ is an orthogonal matrix with $\det(N_g) = 1_s$. We know that due to orthogonality, $T_g$ preserves the canonical inner product on $\mathrm{Cl}(n)$. We define this inner product as $\langle X, Y \rangle = \left\langle X^\dagger Y \right\rangle_0$, where $\langle \cdot \rangle_0$ denotes the scalar part of a multivector.

We will first show that $g^\dagger g = 1_s$. Since $T_g$ is an orthogonal transformation w.r.t. $\langle X, Y \rangle = \left\langle X^\dagger Y \right\rangle_0$, it must satisfy $T_g^* T_g = \mathrm{I}$, where I is the identity map and $T_g^*$ is the adjoint of $T_g$ w.r.t. the inner product. The adjoint $T_g^*$ is given by $T_g^*(Y) = g^\dagger Y g$. This can be verified by checking the main property of an adjoint, $\langle Y, T_g(X) \rangle = \langle T_g^*(Y), X \rangle$:

$$\begin{aligned}
\langle Y, T_g(X) \rangle &= \left\langle Y^\dagger \left( gXg^\dagger \right) \right\rangle_0 \\
&= \left\langle \left( g^\dagger Y^\dagger g \right) X \right\rangle_0 \quad \text{(cyclically permute multivectors inside} \\
&\qquad\qquad\qquad\qquad \text{scalar part operation, } \langle MNP \rangle_0 = \langle PMN \rangle_0) \\
&= \left\langle \left( g^\dagger Y g \right)^\dagger X \right\rangle_0 \quad \text{(since } (ABC)^\dagger = C^\dagger B^\dagger A^\dagger \text{ and } \left( A^\dagger \right)^\dagger = A) \\
&= \left\langle T_g^*(Y), X \right\rangle.
\end{aligned}$$

Now, applying the condition $T_g^* T_g = \mathrm{I}$ means $T_g^* \left( T_g(X) \right) = X$ for all $X \in \mathrm{Cl}(n)$. Substituting the explicit forms for $T_g$ and $T_g^*$, we get

$$g^\dagger \left( gXg^\dagger \right) g = X$$
$$\left( g^\dagger g \right) X \left( g^\dagger g \right) = X.$$

Let $A = g^\dagger g$. The equation becomes $AXA = X$ for all $X \in \mathrm{Cl}(n)$. By the Wedderburn-Artin theorem Cohn [2003], the Clifford algebra $\mathrm{Cl}(n, 0)$ is isomorphic to a matrix algebra (or a direct

sum of two such algebras) over $\mathbb{R}$, $\mathbb{C}$, or $\mathbb{H}$. In such matrix algebras (and hence in $\mathrm{Cl}(n,0)$ or its relevant simple components where $A$ lives, noting $A$ is even), if an element $A$ satisfies $AXA = X$ for all elements $X$ of the algebra, then $A$ must be a scalar multiple of the identity, specifically $\pm 1_s$. Therefore, we must have $A = g^\dagger g = \pm 1_s$.

Fix an orthonormal basis $e_J$ (where $J$ is an ordered multi-index). Note that $e_J^\dagger e_J = 1_s$ for every $e_J$. Write $g = \sum_J c_J e_J$ for scalar coefficients $c_J \in \mathbb{R}$. Then, the scalar part of $g^\dagger g$ is $\langle g^\dagger g \rangle_0 = \sum_J c_J^2$. Since $g$ is invertible, we have $g \neq 0$, so at least one $c_J \neq 0$, making this sum strictly positive. Since $g^\dagger g = \lambda \cdot 1_s$ (where $\lambda = \pm 1_s$), its scalar part is $\lambda$. As this scalar part $\langle g^\dagger g \rangle_0$ must be positive, we must conclude that $\lambda = 1_s$, so $g^\dagger g = 1_s$.

Next, we will show that $gg^\dagger = 1_s$. From the previous result, $g^\dagger g = 1_s$. This means $g$ is invertible and its unique inverse is $g^{-1} = g^\dagger$. Then, we can write $gg^\dagger = gg^{-1} = 1_s$. This helps us establish the first condition required for $g$ to be an element of $\mathrm{Spin}(n)$.

The block-diagonal structure of $\bigoplus_{k=0}^n \mathbb{R}^{\binom{n}{k} \times \binom{n}{k}}$ ensures that the matrix representation $\phi(g)$ acts independently on each grade subspace $\mathrm{Cl}^k(n)$. Consequently, the induced inner automorphism $T_g(X) = gXg^\dagger$ is grade-preserving, mapping each $k$-vector space to itself.

Finally, we will need to show that $g \in \mathrm{Cl}^+(n)$ (i.e., $g$ is an even multivector). Since $N_g \in \mathrm{SO}(2^n)$, we have $\det(N_g) = 1_s$. Since $gg^\dagger = 1_s$, it follows that $g^\dagger = g^{-1}$. The transformation can now be written as $T_g(X) = gXg^{-1}$, which is an inner automorphism of $\mathrm{Cl}(n)$ induced by $g$; such transformations are algebra automorphisms, meaning that they preserve the algebraic product structure (i.e., $T_g(XY) = T_g(X)T_g(Y)$). Since $g$ preserves grade (and in particular preserves the vector subspace $\mathrm{Cl}^1(n)$), the element $g$ can be expressed as a product of invertible vectors. Combined with $gg^\dagger = 1_s$, this means $g$ is a unit versor, i.e., $g \in \mathrm{Pin}(n)$. Versors who are products of an odd number of vectors (odd versors) are elements of $\mathrm{Pin}(n) \setminus \mathrm{Spin}(n)$, while those that are products of an even number of vectors (even versors) are elements of $\mathrm{Spin}(n)$. Note that $\mathrm{Pin}(n)$ is the group of all such versors, and $\mathrm{Spin}(n) = \mathrm{Pin}(n) \cap \mathrm{Cl}^+(n)$.

We know that if $g$ were indeed an odd versor (an element of $\mathrm{Pin}(n) \setminus \mathrm{Spin}(n)$), the induced inner automorphism $X \mapsto gXg^{-1}$ when restricted to the vector subspace $\mathrm{Cl}^1(n)$ is an orthogonal transformation with determinant $-1_s$ (as it includes a reflection, reversing orientation). It is a known result from the representation theory of Clifford algebras that the determinant of the full automorphism $N_g : X \mapsto gXg^{-1}$ acting on the entire algebra $\mathrm{Cl}(n)$ corresponds to the parity of the versor $g$. Specifically, $\det(N_g) = +1_s$ if $g$ is an even versor (i.e., $g \in \mathrm{Spin}(n)$), and $\det(N_g) = -1_s$ if $g$ is an odd versor (i.e., $g \in \mathrm{Pin}(n) \setminus \mathrm{Spin}(n)$). This directly links the parity of the versor $g$ to the determinant of the full transformation matrix $N_g$. Since we know $\det(N_g) = 1_s$, $g$ must be an even versor, i.e., $g \in \mathrm{Cl}^+(n)$.

Since we have shown $gg^\dagger = 1_s$, $g \in \mathrm{Cl}^+(n)$, and that $g$ is grade preserving, we conclude that $g \in \mathrm{Spin}(n)$ by its definition. Thus, $\mathrm{SO}(2^n) \cap \phi\left(\mathrm{Cl}^\times(n)\right) \cap \left(\bigoplus_{k=0}^n \mathbb{R}^{\binom{n}{k} \times \binom{n}{k}}\right) \subseteq \phi(\mathrm{Spin}(n))$. $\qquad\square$

Prop. 3 provides an alternative characterization of the group of matrices associated with rotor conjugation, i.e., $N_r$ lives in the intersection of the special orthogonal group, the subgroup of matrices defined by the group of units of $\mathrm{Cl}(n)$, and those that are grade preserving.

# C  Hyperparameters and Experiment Details

We detail the training configurations for all experiments involving Rotor, Low-Rank (LR), and Block-Hadamard (BH) projections used to approximate attention layers.

## C.1  Training Details

**Learning approximation layers in attention.**   To approximate the linear projections in attention layers of an LLM, we train all replacement modules (Rotor, LR, or BH) by minimizing mean squared error (MSE) loss between the predicted and true projection outputs, based on latent representations extracted from LLMs.

Formally, let $W \in \mathbb{R}^{d_{\text{out}} \times d_{\text{in}}}$ be the dense projection matrix (i.e., query, key, or value) we want to approximate within a transformer block. Given hidden input $x \in \mathbb{R}^{d_{\text{in}}}$, the projection computes

$y = Wx \in \mathbb{R}^{d_{\text{out}}}$. To train an approximate layer, we collect a dataset $\mathcal{D} = \{(x_i, y_i)\}_{i=1}^{N}$ by prompting the LLM with a set of prompts $\{P_j\}_{j \leq n}$ (e.g., `Arc Challenge`) and extracting the relevant hidden states at the target layer. Since hidden states are available for each token position in a sequence for self-attention, we have $N = nT$, where $T$ is the average prompt length. We then learn an approximate layer $H_\theta$ (rotors, LR, or BH) by minimizing

$$\min_\theta \sum_{i=1}^{N} (H_\theta x_i - y_i)^2,$$

where $\theta$ denotes the set of trainable parameters (e.g., bivector coefficients for rotors). Optimization is done via gradient descent using the Adam optimizer [Kingma and Ba, 2017].

We jointly replace the query, key, and value projections within an attention layer of a transformer block. In our experiments, we replace up to three such attention layers and evaluate the resulting model on downstream tasks of perplexity and accuracy metrics across various prompt datasets. When replacing multiple attention layers, say layers $I < J < K$, we train them sequentially in order: first $I$, then $J$, and finally $K$. For each layer, we first replace all earlier trained layers (e.g., $I$ before $J$), and then extract the input-output data for training the new layer under this modified model. This is to ensure that each replacement layer is trained with respect to the distribution induced by preceding replacements. Also, whenever we replace layer $L$, we retrain the output linear projection $W_o^L$ within the same attention block, using the same MSE and Adam optimizer for consistency.

| Model | Key | Query | Value |
|---|---|---|---|
| `LLaMa-3.2 1B` | $2048 \rightarrow 512$ | $2048 \rightarrow 2048$ | $2048 \rightarrow 512$ |
| `LLaMa-3.2 3B` | $3072 \rightarrow 1024$ | $3072 \rightarrow 3072$ | $3072 \rightarrow 1024$ |
| `Qwen-2.5 1.5B` | $1536 \rightarrow 256$ | $1536 \rightarrow 1536$ | $1536 \rightarrow 256$ |
| `Fox-1.0 1.6B` | $2048 \rightarrow 512$ | $2048 \rightarrow 2048$ | $2048 \rightarrow 512$ |

**Table 5:** Input/output hidden dimensions for key, query, and value projections in a single attention layer of different LLMs.

In Tab. 5, we summarize the input and output hidden dimension for different LLMs.

**Rotor networks.** We define the rotor-based transformation as:

$$\psi_{r,s}(x) \triangleq r x s^\dagger,$$

where $r, s \in \text{Spin}(n)$. Each rotor map is then composed as

$$\psi(x) \triangleq \sigma \left( \{ \psi_{r_{ij}, s_{ij}} \left( x^{I_i} \right) \mid i \in [c_1], j \in [c_2] \} \right),$$

where $\sigma$ is a pooling operator applied over the outputs of individual rotor transformation $\psi_{r_{ij}, s_{ij}}$. More details are given in Subsection 4.2. As discussed, each rotor is parameterized by a small number of bivector coefficients that encode geometric rotations, leading to significantly fewer learnable parameters compared to dense or baseline layers. A rotor layer is constructed by stacking multiple rotor maps in depth and arranging them in parallel across width. For example, with width 2 and depth 3, the layer contains 6 rotor maps, each parameterized independently. In our experiments, to reduce computational cost, we use at most width 2 and depth 3 for a rotor layer, which along with any up and down projection needed that is done by the generalized version in 4.2 sums up to roughly 1000 scalar parameters per projection (i.e., per Q/K/V) layer for `LLaMa-3.2 1B`, compared to $1 - 4M$ in original dense layers.

Each rotor map is followed by a sequence of fixed (i.e., parameter-free) permutations, normalizations, and a nonlinearity. Since rotor sandwich products are grade-preserving (see Section B), we apply fixed permutations to enable interaction across grades and increase expressivity. We found that normalization improves training stability.

Hyperparameters such as depth, width, learning rate, and weight decay are selected via grid search; the final values along with the values we explored are listed in Tab. 6. All Clifford algebraic operations, including exponentiation of simple bivectors and sandwich products, are implemented entirely in PyTorch using the `torch_ga` library Alesiani, which supports differentiation. We modified several methods in this package to reduce memory usage. For example, the original package computes the

| Method | Hyperparameter | Values Explored | Final Value |
|---|---|---|---|
| Rotor | Chunk Size | 1024, 2048, 4096 | 2048 |
| | Depth | 1,2,3 | 1 or 3 |
| | Width | 1,2,3 | 1 or 2 |
| | Nonlinearity | ReLU, PReLU, GELU | PReLU |
| | Normalization | true, false | true |
| | Permutations | true, false | true |
| | Learning rate | $0.001, 0.005, 0.01, 0.05$ | $0.05$ |
| | $\ell^2$ weight decay | $0.001, 0.01, 0.1, 0$ | $0$ |
| | Batch size | 16, 32, 64, 128, 256 | 64 |
| | Cosine annealing | true, false | true |
| Low-Rank (LR) | Learning rate | $0.001, 0.005, 0.01, 0.05$ | $0.01$ |
| | $\ell^2$ weight decay | $0.001, 0.01, 0.1, 0$ | $0$ |
| | Batch size | 16, 32, 64, 128, 256 | 256 |
| | Cosine annealing | true, false | true |
| Block-Hadamard (BH) | Block number | 64 | 64 |
| | Learning rate | $0.001, 0.005, 0.01, 0.05$ | $0.01$ |
| | $\ell^2$ weight decay | $0.001, 0.01, 0.1, 0$ | $0$ |
| | Batch size | 16, 32, 64, 128, 256 | 256 |
| | Cosine annealing | true, false | true |

**Table 6:** Hyperparameter settings used for each method.

geometric product using a very sparse three dimensional Cayley table of shape $2^n \times 2^n \times 2^n$ [Hitzer, 2013]. However, since we only require the geometric product between a pure rotor and a multivector when computing the sandwich product, we discard all but $1 + \binom{n}{2}$ parts of the third dimension, rather than keeping the full $2^n$.

**Low-rank approximations.** We use low-rank (LR) approximation as one of our baselines, following prior work such as Hu et al. [2021]. Given a dense matrix $W \in \mathbb{R}^{d_{\text{out}} \times d_{\text{in}}}$, we approximate it as the product of two lower-dimensional matrices: $X \in \mathbb{R}^{d_{\text{out}} \times r}$ and $Y \in \mathbb{R}^{r \times d_{\text{in}}}$, such that

$$W \approx XY,$$

where $r \ll d_{\text{in}}, d_{\text{out}}$. This decomposition effectively constrains the rank of the approximation to at most $r$, capturing a low-rank subspace of the original operator. Low-rank approximations have been shown to be effective in downstream tasks, especially when applied as additive fine-tuning modules to frozen large pre-trained weights. They are also computationally efficient, requiring only $\mathcal{O}(r(d_{\text{out}} + d_{\text{in}}))$ parameters and operations. In our experiments, we choose $r = 1$ and $r = 4$, and LR1 requires roughly $3 - 5\times$ parameters than rotor layers. Hyperparameters are listed in Tab. 6.

**Block-Hadamard approximations.** We adopt the Block-Hadamard (BH) projection as another baseline. The idea is to alternate Hadamard transforms with learnable block-diagonal matrices, enabling a trade-off between expressivity and efficiency through the block size [Zeng et al., 2023]. Formally, it is defined as

$$W \approx \prod_{i=1}^{m} B_i H,$$

where each $B_i$ is a learnable block-diagonal matrix with block size $b$, and $H$ is a fixed Hadamard transform. This approximation requires only $\mathcal{O}(b \cdot \max(d_{\text{in}}, d_{\text{out}}))$ parameters, and can be interpreted as analogous to grouped convolutions followed by channel shuffling [Zhang et al., 2017]. The parameter $m$ controls the depth of the transformation. In our experiments, we use $m = 1$ (i.e., BH1) to reduce parameter count as much as possible and match the scale of rotor-based models. Even with this minimal configuration, BH1 has 8-40$\times$ more parameters than rotors in the case of LLaMa-3.2 1B. Specifically, we use the approximation $W \approx BH$, where $B$ is a block-diagonal matrix composed

of $n$ rectangular blocks, each of height $h = d_{\text{out}}/n$ and width $w = d_{\text{in}}/n$ for the number $n$ of blocks. Hyperparameters are listed in Tab. 6.

**Hyperparameters and model architecture of `FMNIST` experiments.** We trained both dense and rotor-based MLPs under identical conditions. Each model consists of 2 hidden layers (either rotor or dense), followed by ReLU activations. All dense layers were replaced with rotor layers in our rotor variant, except for the final classification head. The dense layers had hidden dimension of 512, while the rotor layers used $\text{Cl}(15)$ with width 3 and depth 1. We performed a search over learning rates $\eta \in (0.001, 0.1)$ and selected $\eta = 0.005$ for the rotor-based model and $\eta = 0.002$ for the dense baseline based on validation accuracy.

## C.2 Computational Resources for Experiments

The experiments for each `LLaMa-3.2 1B` and `Qwen-2.5 1.5B` each took around $1500$ GPU hours, `Fox-1.0 1.6B` around $1000$ GPU hours, and `LLaMa-3.2 3B` around $500$ GPU hours for a total of around $4500$ GPU hours. This was spread across $8$ NVIDIA A100 PCIe GPUs with $40$ GBs of HBM2 memory. Total time of execution was around $3$ weeks of run-time across all $8$ GPUs.

*A note on the execution time of the rotor gadget.* We have not optimized the current code for speed or memory. At inference time, rotor layers are implemented as dense matrix multiplications. Therefore, its runtime scales with the depth of the rotor layer used in the replacement. In our experiments, we used hyperparameters of at most depth 2 and width 3; with an optimized implementation, this will result in at most $2\times$ slowdown, since width is trivially parallelizable.

It is important to note that this corresponds to a direct implementation—we currently do not leverage the block sparsity structure described in Section B.3, as current software support is quite limited. Custom kernels have recently become available to *batch* matrix-matrix multiplication of different dimensions, such as those available in `cublasGemmGroupedBatchedEx`, but our setting requires vector-matrix multiplication. We found that this is slower than performing the dense implementation, though performance is expected to improve significantly once specialized vector-matrix versions become available. While custom kernels such as `cublasSgemmStridedBatched` support batching matrix-matrix multiplication of the *same* dimension, their adaptation to rotors remains limited: even when applied to the parallel portions of the rotor gadget, they must operate sequentially on each block of the block sparse matrices. As a result, this gives a modest speedup, and there remains substantial room for further optimization.

# D Additional Experimental Results

In Fig. 6, we report the number of iterations required by Alg. 2 to converge within a tolerance of $\epsilon = 10^{-3}$, as a function of the number of gradient updates applied to the rotors (i.e., bivector coefficients). Synthetic data was generated with random input and having output come from the rotation corresponding to a random bivector. Each gradient update step is towards learning that random bivector with MSE loss. As expected, higher-dimensional projections require more iterations to converge. Notably, warm-starting from previously learned singular values significantly reduces the convergence speed across all dimensions, supporting our claim in Section 4.1.

In Tab. 8, 9, and 10, we provide additional experimental results on `Qwen-2.5 1.5B`, `LLaMa-3.2 3B`, and `Fox-1.0 1.6B` [Hu et al., 2025], where a single attention layer is replaced by Rotor, LR1, LR4, or BH1 approximations. In Tab. 11, we provide averages for single and two-layer replacements for `Fox-1-1.6B`. The trends observed in the main paper persist: our rotor-based method consistently matches the baselines with significantly fewer parameters (see Tab. 7). These results further support our central claim: *linear layers can be synthesized from a small number of geometric primitives encoding rotations.*

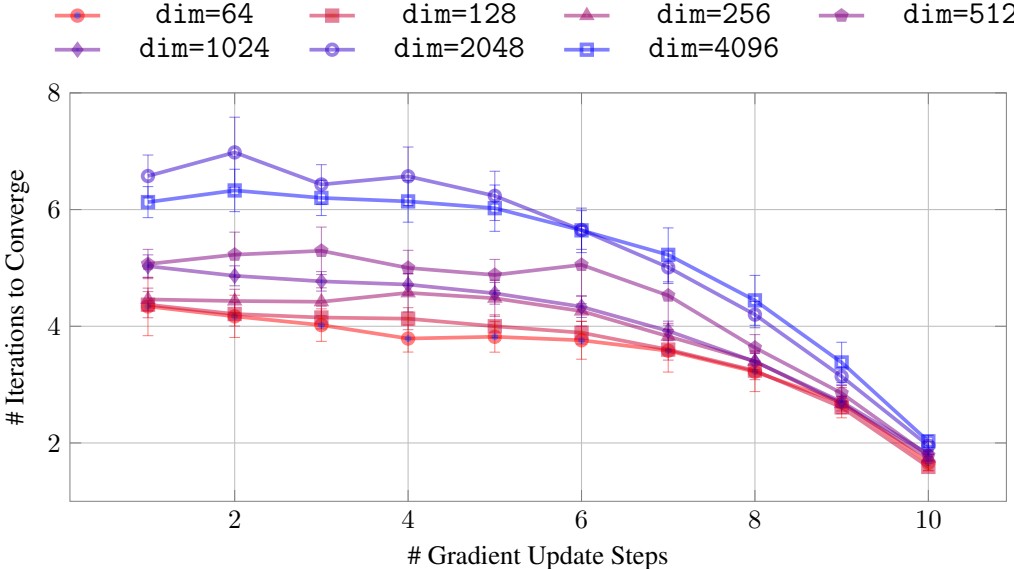

**Figure 6:** Number of iterations required by Alg 2 to converge within a tolerance of $\epsilon = 10^{-3}$, plotted against the number of gradient updates applied to the parameters of rotors (i.e., bivector coefficients). Results are averaged over 50 runs and simple bivectors in the invariant decomposition, with one standard deviation shown as error bars. The results demonstrate that warm-starting with previously learned singular vectors significantly accelerates convergence.

| Model | Method | **Key** | **Query** | **Value** | Total |
|---|---|---|---|---|---|
| | Dense | 1,048,576 | 4,194,304 | 1,048,576 | 6,291,456 |
| | LR1 | 2,560 | 4,096 | 2,560 | 9,216 |
| LLaMa-3.2 1B | LR4 | 10,240 | 16,384 | 10,240 | 36,864 |
| | BH1 | 8,192 | 32,768 | 8,192 | 49,152 |
| | Rotor | ≤1,080 | ≤896 | ≤1,080 | ≤3,056 |
| | Dense | 3,145,728 | 9,437,184 | 3,145,728 | 15,728,640 |
| | LR1 | 4,098 | 6,148 | 4,098 | 14,344 |
| LLaMa-3.2 3B | LR4 | 16,384 | 24,576 | 16,384 | 57,344 |
| | BH1 | 32,768 | 98,304 | 32,768 | 163,840 |
| | Rotor | ≤1,080 | ≤1,120 | ≤1,080 | ≤3,280 |
| | Dense | 393,216 | 2,359,296 | 393,216 | 3,145,728 |
| | LR1 | 1,792 | 3,072 | 1,792 | 6,656 |
| Qwen-2.5 1.5B | LR4 | 7,168 | 12,288 | 7,168 | 26,624 |
| | BH1 | 4,096 | 24,576 | 4,096 | 32,768 |
| | Rotor | ≤904 | ≤896 | ≤904 | ≤2,704 |
| | Dense | 1,048,576 | 4,194,304 | 1,048,576 | 6,291,456 |
| | LR1 | 2,560 | 4,096 | 2,560 | 9,216 |
| Fox-1.0 1.6B | LR4 | 10,240 | 16,384 | 10,240 | 36,864 |
| | BH1 | 8,192 | 32,768 | 8,192 | 49,152 |
| | Rotor | ≤1,080 | ≤896 | ≤1,080 | ≤3,056 |

**Table 7:** Number of parameters for key, query, and value projections in a single attention layer of each model, with the rightmost column showing their sum.

| Dataset | Method | One layer replaced (Layer index) | | | | | | | | | | | | | |
|---|---|---|---|---|---|---|---|---|---|---|---|---|---|---|---|
| | | 1 | 2 | 3 | 4 | 5 | 6 | 7 | 8 | 9 | 10 | 11 | 12 | 13 | 14 |
| Wikitext2 (↓) | LR1 | 2.669 | 2.330 | 2.351 | 2.354 | 2.332 | 2.316 | 2.350 | 2.340 | 2.326 | 2.315 | 2.381 | 2.311 | 2.514 | 2.306 |
| | LR4 | 2.635 | 2.320 | 2.334 | 2.343 | 2.320 | 2.310 | 2.335 | 2.331 | 2.318 | 2.310 | 2.368 | 2.307 | 2.456 | 2.306 |
| | BH1 | 2.341 | 2.316 | 2.329 | 2.323 | 2.323 | 2.308 | 2.338 | 2.321 | 2.314 | 2.307 | 2.354 | 2.308 | 2.417 | 2.307 |
| | Rotor | 2.312 | 2.299 | 2.305 | 2.312 | 2.308 | 2.298 | 2.301 | 2.315 | 2.303 | 2.300 | 2.321 | 2.304 | 2.320 | 2.301 |
| C4 (↓) | LR1 | 3.129 | 2.880 | 2.891 | 2.867 | 2.863 | 2.846 | 2.875 | 2.862 | 2.870 | 2.854 | 2.891 | 2.854 | 2.992 | 2.848 |
| | LR4 | 3.007 | 2.880 | 2.890 | 2.869 | 2.862 | 2.846 | 2.873 | 2.862 | 2.867 | 2.854 | 2.888 | 2.853 | 2.939 | 2.848 |
| | BH1 | 2.943 | 2.872 | 2.878 | 2.864 | 2.856 | 2.844 | 2.870 | 2.858 | 2.863 | 2.849 | 2.879 | 2.851 | 2.925 | 2.847 |
| | Rotor | 2.892 | 2.863 | 2.861 | 2.856 | 2.852 | 2.841 | 2.846 | 2.854 | 2.852 | 2.848 | 2.861 | 2.848 | 2.858 | 2.845 |
| PTB (↓) | LR1 | 3.115 | 3.034 | 3.063 | 3.073 | 3.011 | 3.023 | 3.081 | 3.031 | 3.033 | 3.023 | 3.102 | 3.019 | 3.173 | 3.018 |
| | LR4 | 3.110 | 3.021 | 3.063 | 3.055 | 3.006 | 3.015 | 3.076 | 3.030 | 3.028 | 3.012 | 3.078 | 3.018 | 3.150 | 3.017 |
| | BH1 | 3.091 | 3.007 | 3.046 | 3.044 | 3.004 | 3.007 | 3.076 | 3.020 | 3.022 | 3.007 | 3.064 | 3.015 | 3.101 | 3.012 |
| | Rotor | 3.045 | 2.998 | 3.018 | 3.025 | 3.001 | 2.998 | 3.014 | 3.015 | 3.008 | 3.003 | 3.028 | 3.012 | 3.034 | 3.009 |
| Arc Challenge (↑) | LR1 | 62.23 | 63.95 | 59.23 | 47.21 | 60.52 | 66.52 | 52.36 | 58.37 | 57.51 | 50.21 | 40.34 | 58.80 | 22.32 | 57.51 |
| | LR4 | 59.66 | 62.66 | 64.81 | 54.94 | 60.94 | 66.09 | 56.22 | 62.66 | 66.09 | 58.37 | 55.79 | 61.37 | 9.01 | 58.80 |
| | BH1 | 63.52 | 64.38 | 58.80 | 60.04 | 62.23 | 63.09 | 62.66 | 63.52 | 65.24 | 60.94 | 58.88 | 59.66 | 53.22 | 50.21 |
| | Rotor | 64.81 | 67.38 | 65.61 | 62.23 | 62.66 | 64.38 | 61.80 | 63.95 | 65.24 | 66.09 | 62.23 | 64.81 | 60.09 | 56.22 |
| HellaSwag (↑) | LR1 | 36.00 | 47.00 | 45.00 | 46.67 | 50.00 | 51.00 | 50.33 | 44.00 | 17.33 | 36.00 | 46.00 | 52.67 | 31.33 | 20.00 |
| | LR4 | 38.00 | 54.00 | 46.33 | 48.00 | 51.00 | 55.00 | 56.33 | 51.00 | 26.00 | 43.67 | 49.00 | 54.33 | 31.33 | 20.00 |
| | BH1 | 45.67 | 48.00 | 46.00 | 48.33 | 48.67 | 50.67 | 51.67 | 52.33 | 38.33 | 44.67 | 48.00 | 51.33 | 24.33 | 32.67 |
| | Rotor | 50.67 | 53.00 | 53.67 | 52.33 | 52.00 | 55.33 | 57.00 | 54.67 | 48.67 | 55.33 | 51.00 | 57.00 | 51.33 | 32.00 |

| Dataset | Method | One layer replaced (Layer index) | | | | | | | | | | | | |
|---|---|---|---|---|---|---|---|---|---|---|---|---|---|---|
| | | 15 | 16 | 17 | 18 | 19 | 20 | 21 | 22 | 23 | 24 | 25 | 26 | 27 |
| Wikitext2 (↓) | LR1 | 2.317 | 2.304 | 2.314 | 2.317 | 2.323 | 2.316 | 2.331 | 2.316 | 2.309 | 2.301 | 2.409 | 2.300 | 2.316 |
| | LR4 | 2.317 | 2.304 | 2.314 | 2.317 | 2.323 | 2.316 | 2.331 | 2.316 | 2.309 | 2.301 | 2.409 | 2.300 | 2.316 |
| | BH1 | 2.313 | 2.301 | 2.314 | 2.316 | 2.320 | 2.311 | 2.311 | 2.315 | 2.309 | 2.300 | 2.398 | 2.297 | 2.309 |
| | Rotor | 2.307 | 2.299 | 2.302 | 2.309 | 2.308 | 2.307 | 2.306 | 2.311 | 2.304 | 2.296 | 2.358 | 2.295 | 2.308 |
| C4 (↓) | LR1 | 2.866 | 2.854 | 2.900 | 2.860 | 2.869 | 2.866 | 2.898 | 2.883 | 2.861 | 2.866 | 2.872 | 2.854 | 2.860 |
| | LR4 | 2.865 | 2.851 | 2.899 | 2.859 | 2.868 | 2.866 | 2.889 | 2.866 | 2.856 | 2.851 | 2.858 | 2.853 | 2.858 |
| | BH1 | 2.862 | 2.847 | 2.856 | 2.858 | 2.867 | 2.862 | 2.872 | 2.857 | 2.854 | 2.849 | 2.852 | 2.850 | 2.853 |
| | Rotor | 2.855 | 2.844 | 2.849 | 2.851 | 2.863 | 2.858 | 2.862 | 2.852 | 2.851 | 2.846 | 2.851 | 2.849 | 2.853 |
| PTB (↓) | LR1 | 3.025 | 3.003 | 3.020 | 3.021 | 3.036 | 3.019 | 3.086 | 3.010 | 3.014 | 3.012 | 3.019 | 3.061 | 3.098 |
| | LR4 | 3.024 | 3.003 | 3.026 | 3.018 | 3.034 | 3.017 | 3.017 | 3.010 | 3.015 | 2.997 | 3.017 | 3.012 | 3.024 |
| | BH1 | 3.017 | 3.000 | 3.008 | 3.014 | 3.029 | 3.012 | 3.010 | 3.009 | 3.011 | 2.995 | 3.015 | 3.005 | 3.016 |
| | Rotor | 3.009 | 2.998 | 3.003 | 3.006 | 3.019 | 3.008 | 3.008 | 3.001 | 3.007 | 2.993 | 3.012 | 3.001 | 3.011 |
| Arc Challenge (↑) | LR1 | 41.20 | 65.24 | 27.90 | 65.24 | 68.24 | 16.31 | 47.64 | 67.38 | 66.95 | 65.67 | 65.24 | 66.09 | 66.52 |
| | LR4 | 31.76 | 69.1 | 58.37 | 65.24 | 68.24 | 16.31 | 47.64 | 67.38 | 66.95 | 65.67 | 65.24 | 66.09 | 66.52 |
| | BH1 | 49.79 | 68.67 | 48.50 | 62.23 | 62.24 | 42.49 | 48.93 | 65.24 | 67.81 | 64.39 | 65.31 | 65.79 | 65.24 |
| | Rotor | 56.65 | 70.82 | 61.37 | 60.09 | 69.53 | 51.07 | 47.21 | 67.81 | 66.52 | 66.09 | 65.67 | 66.09 | 65.24 |
| HellaSwag (↑) | LR1 | 28.67 | 49.67 | 32.67 | 45.67 | 51.00 | 5.67 | 36.33 | 52.67 | 57.33 | 53.33 | 53.67 | 53.67 | 54.67 |
| | LR4 | 28.67 | 49.67 | 32.67 | 45.67 | 51.00 | 5.67 | 36.33 | 52.67 | 57.33 | 53.33 | 53.67 | 53.67 | 54.67 |
| | BH1 | 33.67 | 56.67 | 45.67 | 45.67 | 53.33 | 22.67 | 37.67 | 53.67 | 53.67 | 47.67 | 52.33 | 56.00 | 51.67 |
| | Rotor | 42.67 | 55.00 | 49.33 | 51.33 | 52.67 | 18.67 | 35.00 | 51.00 | 56.67 | 54.00 | 54.33 | 54.00 | 56.67 |

**Table 8:** Performance on log-PPL (↓) and accuracy (↑) when replacing **one attention layer** for layer indices 1–27 of `Qwen-2.5 1.5B`. Methods are Low-Rank ($r = 1$ and 4), BH1, and Rotor. Original log-PPL and accuracy are: `Wikitext2` 2.287, `C4` 2.834, `PTB` 2.985, `Arc Challenge` 66.09, `Hellaswag` 55.00.

| Dataset | Method | One layer replaced (Layer index) | | | | | | | | | | | | | |
|---|---|---|---|---|---|---|---|---|---|---|---|---|---|---|---|
| | | 1 | 2 | 3 | 4 | 5 | 6 | 7 | 8 | 9 | 10 | 11 | 12 | 13 | 14 |
| Wikitext2 (↓) | LR1 | 3.867 | 2.539 | 2.525 | 2.521 | 2.517 | 2.509 | 2.557 | 2.499 | 2.507 | 2.519 | 2.510 | 2.505 | 2.550 | 2.519 |
| | LR4 | 3.389 | 2.514 | 2.512 | 2.501 | 2.510 | 2.492 | 2.531 | 2.488 | 2.494 | 2.497 | 2.501 | 2.500 | 2.510 | 2.503 |
| | BH1 | 2.552 | 2.505 | 2.492 | 2.502 | 2.502 | 2.492 | 2.516 | 2.487 | 2.486 | 2.497 | 2.493 | 2.488 | 2.505 | 2.494 |
| | Rotor | 2.594 | 2.520 | 2.497 | 2.499 | 2.517 | 2.498 | 2.521 | 2.488 | 2.492 | 2.511 | 2.499 | 2.492 | 2.506 | 2.510 |

| Dataset | Method | One layer replaced (Layer index) | | | | | | | | | | | | |
|---|---|---|---|---|---|---|---|---|---|---|---|---|---|---|
| | | 15 | 16 | 17 | 18 | 19 | 20 | 21 | 22 | 23 | 24 | 25 | 26 | 27 |
| Wikitext2 (↓) | LR1 | 2.501 | 2.487 | 2.481 | 2.473 | 2.487 | 2.484 | 2.477 | 2.479 | 2.487 | 2.489 | 2.490 | 2.479 | 2.519 |
| | LR4 | 2.497 | 2.485 | 2.477 | 2.471 | 2.480 | 2.479 | 2.471 | 2.467 | 2.480 | 2.474 | 2.471 | 2.471 | 2.503 |
| | BH1 | 2.484 | 2.479 | 2.476 | 2.471 | 2.472 | 2.475 | 2.466 | 2.469 | 2.477 | 2.476 | 2.475 | 2.469 | 2.482 |
| | Rotor | 2.488 | 2.481 | 2.477 | 2.471 | 2.476 | 2.478 | 2.466 | 2.471 | 2.482 | 2.484 | 2.482 | 2.477 | 2.496 |

**Table 9:** Performance on log-PPL (↓) and accuracy (↑) when replacing **one attention layer** for layer indices 1–27 of `LLaMa-3.2 3B`. Methods are Low-Rank ($r = 1$ and 4), BH1, and Rotor. Original log-PPL is 2.460.

| Dataset | Method | One layer replaced (Layer index) | | | | | | | | | | | | | | | |
|---|---|---|---|---|---|---|---|---|---|---|---|---|---|---|---|---|---|
| | | 1 | 2 | 3 | 4 | 5 | 6 | 7 | 8 | 9 | 10 | 11 | 12 | 13 | 14 | 15 | 16 |
| Wikitext2 (↓) | LR1 | 2.645 | 2.573 | 2.549 | 2.535 | 2.531 | 2.546 | 2.550 | 2.533 | 2.581 | 2.540 | 2.557 | 2.546 | 2.538 | 2.538 | 2.536 | 2.546 |
| | LR4 | 2.614 | 2.568 | 2.538 | 2.535 | 2.530 | 2.544 | 2.545 | 2.531 | 2.570 | 2.538 | 2.552 | 2.535 | 2.538 | 2.538 | 2.536 | 2.534 |
| | BH1 | 2.633 | 2.548 | 2.545 | 2.534 | 2.530 | 2.545 | 2.544 | 2.531 | 2.577 | 2.536 | 2.550 | 2.538 | 2.537 | 2.537 | 2.534 | 2.532 |
| | Rotor | 2.544 | 2.542 | 2.541 | 2.532 | 2.529 | 2.538 | 2.540 | 2.530 | 2.537 | 2.533 | 2.549 | 2.536 | 2.534 | 2.531 | 2.531 | 2.531 |
| C4 (↓) | LR1 | 2.997 | 2.915 | 2.879 | 2.877 | 2.877 | 2.891 | 2.884 | 2.878 | 2.889 | 2.886 | 2.876 | 2.884 | 2.874 | 2.881 | 2.878 | 2.875 |
| | LR4 | 2.960 | 2.914 | 2.880 | 2.877 | 2.877 | 2.891 | 2.884 | 2.878 | 2.888 | 2.885 | 2.877 | 2.883 | 2.874 | 2.881 | 2.878 | 2.874 |
| | BH1 | 2.991 | 2.901 | 2.875 | 2.878 | 2.876 | 2.891 | 2.881 | 2.878 | 2.889 | 2.879 | 2.874 | 2.876 | 2.872 | 2.879 | 2.877 | 2.872 |
| | Rotor | 2.924 | 2.901 | 2.875 | 2.876 | 2.876 | 2.890 | 2.881 | 2.878 | 2.882 | 2.878 | 2.873 | 2.876 | 2.872 | 2.878 | 2.876 | 2.871 |
| PTB (↓) | LR1 | 3.359 | 3.270 | 3.222 | 3.231 | 3.226 | 3.255 | 3.222 | 3.231 | 3.279 | 3.233 | 3.229 | 3.268 | 3.232 | 3.237 | 3.231 | 3.279 |
| | LR4 | 3.317 | 3.259 | 3.215 | 3.231 | 3.225 | 3.260 | 3.224 | 3.230 | 3.276 | 3.232 | 3.229 | 3.235 | 3.230 | 3.242 | 3.229 | 3.226 |
| | BH1 | 3.318 | 3.230 | 3.214 | 3.230 | 3.224 | 3.259 | 3.224 | 3.229 | 3.277 | 3.231 | 3.228 | 3.234 | 3.230 | 3.241 | 3.227 | 3.227 |
| | Rotor | 3.283 | 3.229 | 3.212 | 3.229 | 3.224 | 3.257 | 3.223 | 3.228 | 3.255 | 3.229 | 3.228 | 3.232 | 3.228 | 3.234 | 3.225 | 3.225 |
| Arc Challenge (↑) | LR1 | 31.76 | 31.33 | 33.05 | 42.06 | 42.49 | 47.64 | 36.48 | 44.64 | 22.75 | 41.63 | 24.03 | 39.48 | 39.91 | 20.17 | 21.03 | 29.18 |
| | LR4 | 27.47 | 26.61 | 27.90 | 36.91 | 39.06 | 48.50 | 36.05 | 41.20 | 15.45 | 35.19 | 18.88 | 31.33 | 42.06 | 16.74 | 21.03 | 23.61 |
| | BH1 | 34.33 | 21.46 | 28.33 | 32.19 | 38.20 | 45.06 | 26.61 | 42.49 | 11.59 | 36.48 | 14.16 | 21.89 | 46.78 | 24.89 | 25.75 | 9.44 |
| | Rotor | 27.04 | 21.03 | 26.18 | 23.61 | 30.47 | 36.48 | 29.18 | 39.91 | 22.32 | 31.33 | 13.30 | 22.32 | 31.76 | 28.33 | 24.46 | 18.45 |
| HellaSwag (↑) | LR1 | 31.33 | 37.00 | 35.33 | 34.33 | 44.00 | 44.67 | 42.67 | 45.33 | 8.33 | 17.33 | 5.00 | 28.67 | 31.33 | 42.67 | 23.33 | 39.00 |
| | LR4 | 35.67 | 36.67 | 25.00 | 33.00 | 42.67 | 43.33 | 43.00 | 44.67 | 6.33 | 19.00 | 4.33 | 26.67 | 31.33 | 41.33 | 18.67 | 39.33 |
| | BH1 | 40.00 | 34.00 | 24.67 | 35.33 | 44.33 | 45.00 | 42.67 | 45.33 | 9.67 | 19.67 | 5.33 | 22.67 | 36.00 | 40.00 | 29.67 | 33.33 |
| | Rotor | 40.00 | 37.33 | 28.33 | 36.67 | 44.67 | 44.33 | 41.00 | 44.33 | 17.33 | 32.33 | 18.33 | 35.67 | 34.00 | 42.67 | 26.33 | 38.33 |

| Dataset | Method | One layer replaced (Layer index) | | | | | | | | | | | | | | |
|---|---|---|---|---|---|---|---|---|---|---|---|---|---|---|---|---|
| | | 17 | 18 | 19 | 20 | 21 | 22 | 23 | 24 | 25 | 26 | 27 | 28 | 29 | 30 | 31 |
| Wikitext2 (↓) | LR1 | 2.536 | 2.530 | 2.564 | 2.553 | 2.534 | 2.544 | 2.546 | 2.523 | 2.537 | 2.522 | 2.535 | 2.617 | 2.540 | 2.544 | 2.571 |
| | LR4 | 2.534 | 2.528 | 2.563 | 2.553 | 2.534 | 2.543 | 2.541 | 2.523 | 2.536 | 2.523 | 2.534 | 2.558 | 2.531 | 2.543 | 2.541 |
| | BH1 | 2.532 | 2.528 | 2.558 | 2.550 | 2.530 | 2.538 | 2.537 | 2.521 | 2.530 | 2.521 | 2.532 | 2.556 | 2.529 | 2.539 | 2.532 |
| | Rotor | 2.531 | 2.527 | 2.531 | 2.548 | 2.529 | 2.538 | 2.534 | 2.521 | 2.529 | 2.521 | 2.531 | 2.553 | 2.528 | 2.537 | 2.531 |
| C4 (↓) | LR1 | 2.879 | 2.878 | 2.891 | 2.884 | 2.879 | 2.888 | 2.879 | 2.872 | 2.885 | 2.872 | 2.875 | 2.875 | 2.870 | 2.876 | 2.904 |
| | LR4 | 2.879 | 2.877 | 2.891 | 2.884 | 2.880 | 2.889 | 2.879 | 2.872 | 2.884 | 2.872 | 2.874 | 2.875 | 2.870 | 2.875 | 2.890 |
| | BH1 | 2.878 | 2.876 | 2.883 | 2.883 | 2.878 | 2.886 | 2.878 | 2.871 | 2.879 | 2.870 | 2.872 | 2.873 | 2.868 | 2.874 | 2.886 |
| | Rotor | 2.877 | 2.875 | 2.881 | 2.883 | 2.878 | 2.885 | 2.877 | 2.871 | 2.879 | 2.870 | 2.872 | 2.874 | 2.868 | 2.874 | 2.883 |
| PTB (↓) | LR1 | 3.229 | 3.233 | 3.253 | 3.246 | 3.220 | 3.237 | 3.227 | 3.218 | 3.248 | 3.221 | 3.219 | 3.249 | 3.220 | 3.242 | 3.264 |
| | LR4 | 3.224 | 3.221 | 3.239 | 3.240 | 3.214 | 3.226 | 3.222 | 3.213 | 3.230 | 3.219 | 3.218 | 3.222 | 3.212 | 3.233 | 3.235 |
| | BH1 | 3.222 | 3.222 | 3.238 | 3.239 | 3.214 | 3.225 | 3.222 | 3.210 | 3.230 | 3.217 | 3.217 | 3.222 | 3.212 | 3.221 | 3.226 |
| | Rotor | 3.221 | 3.220 | 3.232 | 3.238 | 3.213 | 3.225 | 3.220 | 3.210 | 3.228 | 3.216 | 3.217 | 3.220 | 3.211 | 3.220 | 3.223 |
| Arc Challenge (↑) | LR1 | 4.72 | 6.87 | 39.91 | 44.21 | 18.45 | 33.91 | 34.33 | 21.46 | 7.30 | 27.47 | 18.88 | 29.18 | 47.64 | 10.30 | 7.73 |
| | LR4 | 6.87 | 8.15 | 38.63 | 41.63 | 18.88 | 30.90 | 34.33 | 21.46 | 8.58 | 27.04 | 18.88 | 31.76 | 48.07 | 12.45 | 13.73 |
| | BH1 | 10.30 | 5.58 | 35.62 | 42.49 | 20.17 | 30.04 | 32.19 | 24.03 | 9.87 | 25.32 | 19.31 | 25.32 | 46.78 | 18.88 | 24.46 |
| | Rotor | 20.17 | 18.88 | 25.32 | 33.91 | 21.03 | 30.04 | 26.61 | 24.46 | 11.59 | 26.18 | 21.46 | 24.89 | 45.49 | 19.74 | 24.46 |
| HellaSwag (↑) | LR1 | 27.33 | 26.33 | 42.33 | 47.33 | 41.33 | 37.00 | 37.67 | 32.00 | 23.67 | 40.00 | 34.67 | 46.67 | 47.33 | 28.67 | 12.00 |
| | LR4 | 25.00 | 28.33 | 41.00 | 46.33 | 40.00 | 37.00 | 38.67 | 32.33 | 26.67 | 40.67 | 36.33 | 44.33 | 46.00 | 31.33 | 28.00 |
| | BH1 | 28.67 | 27.67 | 42.33 | 47.67 | 44.00 | 40.67 | 38.33 | 34.67 | 27.33 | 39.67 | 36.67 | 42.33 | 47.00 | 32.33 | 36.67 |
| | Rotor | 33.67 | 34.00 | 33.00 | 34.67 | 37.67 | 41.00 | 38.33 | 34.33 | 29.67 | 39.67 | 33.67 | 40.67 | 46.33 | 31.67 | 38.67 |

**Table 10:** Performance on log-PPL (↓) and accuracy (↑) when replacing **one attention layer** for layer indices 1–31 results of `Fox-1.0 1.6B`. Methods are Low-Rank ($r = 1$ and $4$), BH1, and Rotor. Original log-PPL and accuracy are: `Wikitext2` 2.517, `C4` 2.862, `PTB` 3.205, `Arc Challenge` 24.89, `Hellaswag` 38.33.

| | Dataset | Method | Fox-1.0 1.6B | |
|---|---|---|---|---|
| | | | one | two |
| Log-PPL | Wikitext2 | Original | —— 2.517 —— | |
| | | LR1 | 2.550 | 2.589 |
| | | LR4 | 2.540 | 2.578 |
| | | BH1 | 2.538 | 2.573 |
| | | Rotor | 2.534 | 2.576 |
| | C4 | Original | —— 2.862 —— | |
| | | LR1 | 2.881 | 2.907 |
| | | LR4 | 2.880 | 2.901 |
| | | BH1 | 2.878 | 2.898 |
| | | Rotor | 2.877 | 2.901 |
| | PTB | Original | —— 3.205 —— | |
| | | LR1 | 3.238 | 3.299 |
| | | LR4 | 3.230 | 3.276 |
| | | BH1 | 3.228 | 3.275 |
| | | Rotor | 3.226 | 3.259 |
| Accuracy (%) | Arc Challenge | Original | —— 24.89 —— | |
| | | LR1 | 29.03 | 26.40 |
| | | LR4 | 27.40 | 27.12 |
| | | BH1 | 26.77 | 31.42 |
| | | Rotor | 25.82 | 30.22 |
| | Hellaswag | Original | —— 38.33 —— | |
| | | LR1 | 32.33 | 25.49 |
| | | LR4 | 32.41 | 24.40 |
| | | BH1 | 33.92 | 27.73 |
| | | Rotor | 35.14 | 32.87 |

**Table 11:** Log-PPL (↓) and accuracy (↑) using original, Low-Rank ($r = 1$ or $4$), BH1, and Rotor for 1–2 layer replacements on Fox-1.0 1.6B. One-layer results are averaged over all layers; two-layer results are averaged over five random selections. Red indicates best, blue second-best per setting.

