# OpenReview forum: "Composing Linear Layers from Irreducibles"
_NeurIPS.cc/2025/Conference — NeurIPS 2025 poster_

### Official Review · Reviewer_dZ69 · 2025-06-20

**Clarity:** 3
**Significance:** 3
**Originality:** 4
**Rating:** 5
**Confidence:** 4

**Summary:**

This paper presents a method for synthesizing linear layers from a small number of geometric primitives (irreducibles) encoding rotations.
For this, the authors utilize the framework of Clifford (geometric) algebra and introduce a differentiable algorithm that
decomposes them into products of rotors.
The proposed construction, in which linear layers can be expressed as compositions of bivectors, uses only $\mathcal{O}(\log^2 d)$ parameters, significantly reducing the number required by dense matrices — $\mathcal{O}(d^2)$, with $d$ being the input/output dimension.
To demonstrate the effectiveness of the method, the authors use their rotor-based construction and substitute the K, Q, and V dense matrices in LLM attention layers, exhibiting comparable performance (in accuracy and perplexity) to strong baselines such as block-Hadamard and low-rank approximations, across different datasets.
Overall, the paper demonstrates that bivectors in Clifford algebra can effectively compose linear layers.

**Questions:**

Q1. Pertaining to W1 and with the current implementation, how does the computational speed (both training and inference) of the proposed bivector method compare to the related methods (Dense, BH, LR)?

Q2. With regards to W2, could the authors run a simple experiment, e.g., an MNIST classification with a feed-forward model, using their layer construction method and the baseline/related ones, training the model from scratch (as opposed to using a pre-trained LLM)?

Q3. Suggestion: It would improve the paper if the limitations were stated in the main paper, or at least, if it were mentioned that they are given in the Appendix.

The final rating depends on the author’s rebuttal.

**Ethical Concerns:**

["NO or VERY MINOR ethics concerns only"]

**Final Justification:**

My questions are well answered, and I will maintain the original score. I strongly recommend that the authors update their manuscript with the responses provided in the rebuttal, to the extent possible. This includes the FMNIST experiment.

**Limitations:**

yes

**Quality:**

3

**Strengths And Weaknesses:**

First of all, I appreciate seeing the powerful toolbox of Clifford algebra being further explored in neural processing. The strengths of this paper are pretty straightforward to identify:

S1. The paper is well-written. The storyline is clear. The claims are supported by experiments in a well-defined scope and framework.

S2. Novelty and significance: The idea of expressing linear transformations as compositions of bivectors using sandwich products results in an exponential reduction of the required number of parameters for linear layers.

S3. Compatibility: The proposed differentiable invariant decomposition algorithm enables integration with autograd and gradient-based optimization.

----
As for the weaknesses, I have identified the following, despite the formulated scope and goals of the experiments and the current implementation limitations clearly stated by the authors:

W1. No comparison of the computational time of the related methods (Dense, BH, LR) is presented.

W2. No experiment in which the proposed bivector-composed linear layers are learned in a network end-to-end, along with a computational cost comparison against dense-matrix linear layers. Such an experiment would serve the paper well in that it would help a broader audience of readers to understand better the scope of the current limitations and the applicability.

W3. Minor:
- Currently, this work is more of a feasibility study.
- Equation 3 is missing its number; besides, all the equations should be numbered for consistency.

---

> ### Author Rebuttal · Authors · 2025-07-30
>
> We thank the reviewer for taking the time to carefully engage with our work, and for the thoughtful suggestions and questions. We answer each question below.
>
> **Q1: Pertaining to W1 and with the current implementation, how does the computational speed (both training and inference) of the proposed bivector method compare to the related methods (Dense, BH, LR)?**
>
> Currently, rotor layers are slower than dense layers in wall-clock inference runtime due to the overhead of the unoptimized Clifford algebra operations. Inference is about $(\text{width} \times \text{depth})/4$ times slower, where width and depth of our rotor gadget are at most 3 and 2 in our experiments. Training has the extra cost of storing the gradients, but the backpropagation through decomposition (see Figure 3) from applied weights to actual parameters, which involves $\log_2 d$ block-sparse matmuls, where $d$ is the model dimension, can be done on a different GPU while backpropagation proceeds through the rest of the network.
>
> **We believe we could not only close this gap, but rotor layers may outperform dense layers**, and here is why. The primary goal of our work was to *synthesize dense linear layers from exponentially fewer geometric primitives.* **The real opportunity unlocked by our geometric compotition lies in eliminating memory bandwidth bottlenecks**, as also described in our answer to Reviewer tqhq. NVIDIA Research's recent LIMINAL paper (arXiv:2507.14397) identifies three critical insights for LLM inference:
>
> 1) Memory capacity requirements are enormous (385GB+ for Llama-405B)
> 2) Memory bandwidth is the primary bottleneck for user tokens/second
> 3) DRAM-based architectures offer fundamental efficiency advantages
>
> The reviewer will see that **our approach directly addresses finding #2**: rather than reading in millions of parameters from memory to GPU (the dominant cost in LLM inference), we can, with enough engineering, generate these weights on-chip by reading only ~1000 geometric parameters and constructing rotors (see Figure 3) on the fly! This trades compute for memory bandwidth—an extremely favorable exchange given that memory, not compute, is the primary constraint as noted in the LIMINAL work and other industry reports.
>
> As an aside, this decomposition reminds us of the Fast Fourier Transform: factoring the DFT matrix didn't immediately reduce runtime, yet this compositional insight later enabled $O(n \log n)$ algorithms compared to $O(n^2)$. Likewise, our construction reveals that dense linear layers have far fewer degrees of freedom than their raw parameter counts suggest, allowing for potential new algorithms to take advantage.
>
> In summary, this direction is highly promising, particularly for scaling up inference efficiency in large models. However, realizing the full benefit would require substantial engineering work such as implementing sparse-aware Clifford libraries and optimizing GPU memory layouts, which is beyond the scope of this paper. For these reasons, we do not compare wall-clock runtime to other baselines.
>
> **Q2: With regards to W2, could the authors run a simple experiment, e.g., an MNIST classification with a feed-forward model, using their layer construction method and the baseline/related ones, training the model from scratch (as opposed to using a pre-trained LLM)?**
>
> Yes, thank you for this excellent suggestion!
>
> We ran additional expeirments with an MLP architecture on FMNIST, where we replace all dense layers (except the final classification head) with rotor layers and trained both end-to-end from scratch. We performed a hyperparameter search for both and trained each for the same number of epochs. In this setting, the conventional dense layers and rotor layers achieved 89.67% and 88.36% accuracy respectively.
>
> As you noted, full end-to-end training on large models like LLMs is currently infeasible due to limitations in Clifford algebra libraries and their hardware integration. These results on FMNIST suggest that rotor layers can be competitive with dense layers when trained from scratch, giving us encouraging evidence for scalability with more engineering effort.
>
> **Suggestion: It would improve the paper if the limitations were stated in the main paper, or at least, if it were mentioned that they are given in the Appendix.**
>
> We have now added a reference to this section in the main paper’s conclusion. Thanks for the suggestion!

---

> > ### Comment · Reviewer_dZ69 · 2025-08-05
> >
> > I thank the authors for their rebuttal. My questions are well answered, and I will maintain the original score.
> > I strongly recommend that the authors update their manuscript with the responses provided in the rebuttal. This includes the FMNIST experiment.

---

> > > ### Author Response · Authors · 2025-08-06
> > > **Follow-up to Comment**
> > >
> > > We thank the reviewer for the follow-up. We will certainly incorporate the new FMNIST experiments, along with the discussion on computational efficiency and future directions, into our revised draft. Thank you again for your helpful feedback.

---

### Official Review · Reviewer_EgXM · 2025-07-02

**Clarity:** 2
**Significance:** 2
**Originality:** 3
**Rating:** 4
**Confidence:** 3

**Summary:**

The paper investigated how linear transformations in neural networks and LLMs can be synthesized from geometric primitives via Clifford algebra.
The authors showed that linear layers can be represented as compositions of rotors, which are parameterized by bivectors (geometric objects encoding oriented planes).
They introduce a differentiable invariant decomposition algorithm that breaks general bivectors into another components with closed-form exponential. The authors replaced the key, query, and value projection layers in the attention mechanism of two LLMs with their proposed layers to evaluate their method.

**Questions:**

Can the authors add more on the runtime of their proposed method and does it compare to the baselines?

**Ethical Concerns:**

["NO or VERY MINOR ethics concerns only"]

**Final Justification:**

The rebuttal addressed some of my questions and I already have a positive score.

**Limitations:**

Yes.

**Paper Formatting Concerns:**

No.

**Quality:**

3

**Strengths And Weaknesses:**

**Strengths**:

- Using Clifford algebra to represent linear layer: The paper provided an algebraic lens on linear layers, showing that they admit a compact factorization into bivector irreducibles and rotors, with less number of parameters.
- Empirical results: The authors showed that replacing the key, query, and value projections in LLaMA-3.2 1B and Qwen-2.5 1.5B with rotor-based layers achieved competitive performance aganist low-rank and block-Hadamard baselines on Wikitext2, C4, and PTB datasets and Arc Challenge & HellaSwag benchmarks.


**Weaknesses**

- Practical Computational Cost: The paper focuses on parameter efficiency, but I think the practical computational cost is an important consideration. The authors mentioned that the current implementation relies on dense matrix representations and does not yet exploit the sparsity of the rotor decomposition.

---

> ### Author Rebuttal · Authors · 2025-07-30
>
> Thank you for the insightful question and positive feedback! We answer it below.
>
> **Q: Can the authors add more on the runtime of their proposed method and does it compare to the baselines?**
>
> Currently, rotor layers are slower than dense layers in wall-clock inference runtime due to the overhead of the unoptimized Clifford algebra operations. Inference is about $(\text{width} \times \text{depth})/4$ times slower, where width and depth of our rotor gadget are at most 3 and 2 in our experiments with minimal use of the structure of our matrices.
>
> **We believe we could not only close this gap, but rotor layers may outperform dense layers**, and here is why. The primary goal of our work was to *synthesize dense linear layers from exponentially fewer geometric primitives.* **The real opportunity unlocked by our geometric compotition lies in eliminating memory bandwidth bottlenecks**, as also described in our answer to Reviewer tqhq. NVIDIA Research's recent LIMINAL paper (arXiv:2507.14397) identifies three critical insights for LLM inference:
>
> 1) Memory capacity requirements are enormous (385GB+ for Llama-405B)
> 2) Memory bandwidth is the primary bottleneck for user tokens/second
> 3) DRAM-based architectures offer fundamental efficiency advantages
>
> The reviewer will see that **our approach directly addresses finding #2**: rather than reading in millions of parameters from memory to GPU (the dominant cost in LLM inference), we can, with enough engineering, generate these weights on-chip by reading only ~1000 geometric parameters and constructing rotors (see Figure 3) on the fly! This trades compute for memory bandwidth—an extremely favorable exchange given that memory, not compute, is the primary constraint as noted in the LIMINAL work and other industry reports.
>
> As an aside, this decomposition reminds us of the Fast Fourier Transform: factoring the DFT matrix didn't immediately reduce runtime, yet this compositional insight later enabled $O(n \log n)$ algorithms compared to $O(n^2)$. Likewise, our construction reveals that dense linear layers have far fewer degrees of freedom than their raw parameter counts suggest, allowing for potential new algorithms to take advantage.
>
> In summary, this direction is highly promising, particularly for scaling up inference efficiency in large models. However, realizing the full benefit would require substantial engineering work such as implementing sparse-aware Clifford libraries and optimizing GPU memory layouts, which is beyond the scope of this paper. For these reasons, we do not compare wall-clock runtime to other baselines.

---

> > ### Comment · Reviewer_EgXM · 2025-08-07
> >
> > I thank the authors for their response. I'll keep the score towards acceptance.

---

> > > ### Author Response · Authors · 2025-08-07
> > > **Follow-up to Comment**
> > >
> > > Thank you for your response. We're happy to hear that our rebuttal was helpful.

---

### Official Review · Reviewer_7rjk · 2025-07-03

**Clarity:** 3
**Significance:** 2
**Originality:** 3
**Rating:** 5
**Confidence:** 2

**Summary:**

The paper posits a way to rewrite linear transformations in deep learning frameworks in a nonlinear algebraic form with fewer parameters, based on composition of simple primitives. These primitives are based on bivector products in Clifford algebra. The expression is differentiable via chain rule and it applies to LLMs (on which experiments are done to compare to other model compression algorithms), and the compression goes from scaling as d^2 of number of learnable parameters to that number scaling like log^2d in which d is the layer dimension.

**Questions:**

I'm impressed by the experimental results, such as 4700x reduction of number of parameters, and I would like to understand possible limitations of the method. So I try to ask some questions in that direction below

1) In theorem 2 and in the LLM layer experiments you have a number of parameters going down by a huge factor. This means that the models become highly compressed by this approach. Does it mean that the learning rate on the compressed models become comparatively slower as the parameters get more entangled? Or else, how do the learning rates compare?

2) How do you explain the fact that your rotor based composition works well with 1 or 2 layers in experiments but then stops improving? (as mentioned in line 318 and following ones)

3) Do you think that the composition of primitives as given by your result can be interpretable in some setting (maybe some LLM case) or in some toy example? You say in line 349-350 that "Our goal is to understand the compositional structure of the linear transformation itself" so can you please state in more detail what do we understand now with your method precisely?

4) Do you know, besides drop in parameters, what is a drop in Kolmogorov Complexity incurred during the learning dynamics by your method compared to others? I'd be curious to know how much information is lost due to gradient descent not being able to fit the models, and I'm wondering if there is a metric or a toy case in which you can give more insights on that, besides computing the accuracy or PPL as you do.

**Ethical Concerns:**

["NO or VERY MINOR ethics concerns only"]

**Final Justification:**

Authors replies cover my doubts and I'm raising score from 4 to 5 to quantify this.

**Limitations:**

yes

**Paper Formatting Concerns:**

No concerns.

**Quality:**

3

**Strengths And Weaknesses:**

Strengths: the method seems interesting and it performs well compared to the presented alternatives, therefore it is a good advance as far as I could check.

I feel that possible limitations of the method are not discussed, which can mean two things: either this is a cornerstone paper which will revolutionize LLM and general model compression, or this lack of discussion of limitations is a weakness of the paper. Unfortunately I can't tell which one is the case, as I have never worked with Clifford algebra and thus this is just marginal to my domain of expertise

---

> ### Author Rebuttal · Authors · 2025-07-30
>
> We thank the reviewer for the insightful questions and positive feedback. Below we respond in detail to all questions.
>
> **Q1: In theorem 2 and in the LLM layer experiments you have a number of parameters going down by a huge factor. This means that the models become highly compressed by this approach. Does it mean that the learning rate on the compressed models become comparatively slower as the parameters get more entangled? Or else, how do the learning rates compare?**
>
> Thank you for the interesting question. Full end-to-end training on LLMs is currently infeasible due to limitations in Clifford algebra packages and hardware integration. To study this question more carefully, we ran additional experiments with a MLP trained on FMNIST, where we replace all dense layers (except the final classification head) with rotor layers. The accuracy as epochs increase are shown below:
>
> **Accuracy (%)**
> | #Epoch | 1 | 2 | 3 | 4 | 5 | 6 | 7 | 8 | 9 | 10 |
> |-|-|-|-|-|-|-|-|-|-|-|
> | Dense  |$85.05$ |$86.23$ |$87.17$ |$88.07$ |$87.90$ |$88.54$ |$89.07$ |$89.36$ |$89.53$ |$89.67$ |
> | Rotor |$80.80$ |$82.05$ |$82.60$ |$82.38$ |$86.14$ |$86.75$ |$86.74$ |$86.52$ |$87.94$ |$88.36$ |
>
> From these experiments, we observed:
>
> 1. Accuracy improves more quickly during training with dense layers than with rotor layers
> 2. Both methods ultimately reach a final accuracy of 88-89%
>
> Your intuition is correct. In this setting, the learning rate on the compressed model is comparatively slower than the dense approach.
>
>
> **Q2: How do you explain the fact that your rotor based composition works well with 1 or 2 layers (depth) in experiments but then stops improving (as mentioned in line 318 and following ones)?**
>
> Thank you for pointing this out! We realize that Figure 5 can be made clearer. Our current setup can be viewed as fitting to a fixed target functionality of the original dense matrix, where we train only to replace that layer while keeping all other weights frozen. This is different from end-to-end training, where more abstract representations across layers co-adapt. Therefore, the best possible perplexity we can reach is that of the original LLM.
>
> From Table 1, we see this lower bound is 9.85. Looking at Figure 5, we see that after a depth of 2, the PPL is already 10.4--somewhat close to the limit, with a depth of 5 reaching 10.1. The PPL doesn't improve as much beyond depth 2 or 3 because the model is already nearing the lower bound. For clarity, we have added a horizontal line at 9.85 in Figure 5 to emphasize this convergence.
>
> This behavior can be understood more precisely as follows. Since we use pre-trained LLMs, the attention projection matrices (i.e., query, key, and value) may already lie close to the subspace representable by our rotor construction. As we are approximating the functionality of fixed matrices rather than training from scratch, 1-2 rotor layers may capture most of the structure needed to match the pre-existing functionality. Additional layers then provide only marginal refinement, rather than letting us access new transformations.
>
>
> **Q3: Do you think that the composition of primitives as given by your result can be interpretable in some setting (maybe some LLM case) or in some toy example? You say in line 349-350 that "Our goal is to understand the compositional structure of the linear transformation itself" so can you please state in more detail what do we understand now with your method precisely?**
>
> Thank you for asking this important question.
>
> First, we now understand that the function space learned by linear layers in LLMs can be represented using objects with $O(\log^2 d)$ geometric degrees of freedom, rather than $O(d^2)$ arbitrary parameters. This suggests that the linear transformations (via dense layers) live near highly structured objects that preserve similar invariances in the data. This gives us a *clearer perspective on their intrinsic geometric simplicity*.
>
> Second, each bivector-rotor pair in our decomposition operates independently on a $2^n$-dimensional subspace. We now understand that the seemingly monolithic $d\times d$ transformation factors into smaller, independent modules--a structure that is very different from low-rank or block-diagonal approximations. This modularity is discovered rather than imposed. As a result, these transformations can be generated from compact geometric primitives instead of being stored as large matrices.
>
> Taken together, this understanding suggests that the underlying computations in LLMs may exhibit hidden geometric simplicity, and that the models could be implicity discovering and exploiting such structure in the data. While we do not yet claim semantic interpretability of the geometric primitives in large models, this can open promising directions for both interpretability research and the design of more efficient architectures. For an example of how this compositional structure can be exploited for efficieny gains, please see our response to Q3 for Reviewer tqhq.
>
>
> **Q4: Do you know, besides drop in parameters, what is a drop in Kolmogorov Complexity incurred during the learning dynamics by your method compared to others? I'd be curious to know how much information is lost due to gradient descent not being able to fit the models, and I'm wondering if there is a metric or a toy case in which you can give more insights on that, besides computing the accuracy or PPL as you do.**
>
> We much appreciate this suggestion to explore the information-theoretic implications of our construction. Our background in this topic is somewhat limited, and certainly not at a level to provide a fully satisfying answer.
>
> From our understanding, Kolmogorov Complexity is upper bounded by the number of parameters. For our general rotor gadget (see Appendix B.2), this corresponds to $2c_1c_2{\log_2 d \choose 2}$, where $c_1$ and $c_2$ are hyperparameters and $d$ is the hidden dimension. For the models used in our experiments, we provide parameter counts of the dense linear layer, rotor layer, and baselines in Table 6, Appendix D.
>
> We would be grateful for any guidance or references to theoretical or empirical results that could help us explore this direction more rigorously. If we are able to formalize a meaningful result, we will include it in the final version and credit the reviewer for the suggestion.

---

> > ### Comment · Reviewer_7rjk · 2025-08-01
> >
> > Thanks for engaging with the questions.
> >
> > I think that Q1 is answered very nicely, and this settles my doubt. For Q2 too, I am satisfied with the reply.
> >
> > For Q3, this does not speak directly of interpretability, you just mention a hint that some kind of codification is underway, and this we can believe without interpreting the layers. What I was (ambitiously) expecting as a reply would have been some direct observation that the learned representations are interpretable. I think this is left for future work/attempts. By the way, I liked a lot the reply to Q3 of reviewer tqhq, this sounds exciting.
> >
> > For Q4, as I said initially "I am wondering if there is a metric or toy case" so I don't know one (otherwise I would have said it earlier!). I guess this is a separate direction for future work, similarly to Q3.
> >
> > In summary, I am satisfied with the responses, and I think that the paper should be accepted. As a way of quantifying that, I'll change score to 5.

---

> > > ### Author Response · Authors · 2025-08-01
> > > **Follow-up to Comment**
> > >
> > > We sincerely thank the reviewer for their thoughtful engagement with our rebuttal. We're glad to hear that our responses were satisfying, and we will incorporate the discussion--particularly on learning rates, rotor gadget depths, and geometric interpretability--into the revised draft. Thank you again for the valuable suggestions and encouraging feedback.

---

### Official Review · Reviewer_tqhq · 2025-07-03

**Clarity:** 4
**Significance:** 3
**Originality:** 3
**Rating:** 4
**Confidence:** 4

**Summary:**

The authors show that replacing linear layers with clifford algebra layers can improve interpretability while maintaining comparable performance and reducing parameter count. To do this, the authors use an existing theoretical result stating that any linear function can be written as a sandwich product with clifford algebra multivectors. Naively designing an architecture based on this result would introduce a huge computational burden. Instead, the authors claim any linear function can be written as the composition of rotors which have many fewer parameters. While this theoretical result is not proved in the main body, it is empirically validated in the case where 1-3 attention layers in a small LLM are replaced with rotor layers.

**Questions:**

*Questions*
- (line 111) It isn’t clear to me how choosing to consider the subalgebra Spin(n) still allows you to represent a general, arbitrary linear map.
- (line 256) Do the authors expect that performance would remain comparable if all layers were replaced with clifford algebra layers?
- (line 287) Are the clifford algebra layers harder to train than conventional linear layers? Are they more computationally expensive?

*Comments*
- It would be easier to read Table 2 if the best performing methods were highlighted as they are in Table 1

**Ethical Concerns:**

["NO or VERY MINOR ethics concerns only"]

**Final Justification:**

My perception of the work was relatively unchanged by the rebuttal, however, I am more confident in my assessment.

**Limitations:**

The authors do not acknowledge limitations or potential negative societal impacts of their work.

**Paper Formatting Concerns:**

No formatting concerns

**Quality:**

3

**Strengths And Weaknesses:**

The submission appears technically sound, with the exception that a main claim (any linear function can be written as a sandwich product with rotors) is not proved. The claim is, however, supported in a narrow context by experimental results.
The submission is very well written and organized.
The results appear significant, potentially providing a mechanism to improve interpretability in LLMs.
The work is part of a growing body of research leveraging clifford algebra representations for interpretability and expressivity in neural networks. As far as I know, the insights presented in this work are original and seem to expand our understanding of the utility of clifford algebra representations.

---

> ### Author Rebuttal · Authors · 2025-07-30
>
> We thank the reviewer for their constructive comments and thoughtful questions. Below we respond in detail to all questions.
>
> **Q1 (line 111): It isn’t clear to me how choosing to consider the subalgebra $\text{Spin}(n)$ still allows you to represent a general, arbitrary linear map.**
>
> Thank you for raising this subtle point. As you noted, we use a known result (Lemma $1$ by Hestenes and Sobczyk) which states that a general linear map can be written as a finite sum of the conjugations of multivectors in $\text{Cl}(n)$. This is useful, but directly using multivectors is computationally expensive. To make this practical, we must impose additional structure on the multivectors that can be exploited. The subset that we choose, the Spin group $\text{Spin}(n)$, does not capture arbitrary maps in $\text{GL}(d)$, but rather *a subset of maps defined by rotors*. In Appendix B.3 (Property 3), we fully characterize the matrices isomorphic to the image of rotor sandwich products as a subgroup of $\text{SO}(2^n)$ induced by the Clifford algebra.
>
> What helps in recovering expressivity is our parallel rotor gadget: using rotor modules acting on different subspaces and aggregating their outputs through pooling, we approximate more general linear maps through superposition. Each rotor contributes an orthogonal transformation on its subspace, and their weighted combination approximates non-orthogonal transformations. While we cannot represent all of $\text{GL}(d)$ exactly (unless we go back to Lemma 1), empirically this is sufficient. We find that this subspace is expressive enough to approximate LLM projection layers.
>
>
> **Q2 (line 256): Do the authors expect that performance would remain comparable if all layers were replaced with clifford algebra layers?**
>
> Yes, for smaller models performance does remain comparable to the dense linear layer. We ran additional experiments on FMNIST, where we trained an MLP end-to-end from scratch, replacing *all* dense layers with rotor layers (except the classification head). We performed a hyperparameter search for both and trained each for the same number of epochs. In this setting, the conventional dense layers and rotor layers achieved 89.67% and 88.36% accuracy respectively.
>
> For large models such as LLMs, full end-to-end training is currently infeasible due to the nascent state of Clifford algebra packages and their limited hardware integration. For this reason, we chose layer-wise replacement, keeping all other weights frozen. When multiple layers are replaced with any approximation method under this scheme, small errors can accumulate across layers.
>
> To our knowledge, there are only two PyTorch-compatible packages that implement Clifford operations. On line 764, we discuss one of the many optimizations we made to the *torch_ga* package (which we provide in the supplementary material). This allows us to execute key Clifford algebra operations (e.g., wedge product, left contraction, sandwich product) for the larger algebras required in LLMs, but significantly more engineering effort is needed to make full-network training efficient at scale.
>
>
> **Q3 (line 287): Are Clifford layers harder to train or more computationally expensive?**
>
> Yes, Clifford (rotor) layers are currently more computationally intensive to train, but more efficient to apply at inference time (with a suitably optimized low-level implementation). For example, on LLaMA-1B and 3B, rotor layers require 35.8M and 82.4M FLOPs (multiplied by a factor $K$ which ranges from 1 to 6 depending on the configuration of rotor gadget), compared to 206.1M and 515.3M FLOPs for standard dense layers. Thus, even in the worst case, it works out favorably.
>
> However, we emphasize that this comparison is not our primary goal. Our core contribution is to **synthesize dense linear layers from exponentially fewer geometric primitives, rather than to minimize FLOPS of linear layers directly.**
>
> The FLOP reduction is coincidental: the synthesized matrices simply happen to have fewer non-zero entries due to their block-diagonal structure. We were not targeting sparsity; in this specific construction, it is a byproduct of our geometric formulation. It is possible that alternative compositional approaches could synthesize the exact dense matrix with no sparsity (and FLOP) benefits.
>
> **The real promise going forward lies in eliminating memory bandwidth bottlenecks**. The recent LIMINAL paper (arXiv:2507.14397) from NVIDIA Research identifies three critical findings for LLM inference:
>
> 1) Memory capacity requirements are massive (385GB+ for Llama-405B)
> 2) Memory bandwidth is the primary bottleneck for user tokens/second
> 3) DRAM-based designs offer fundamental efficiency advantages
>
> The reviewer will see that **our approach directly addresses finding #2**: instead of loading millions of parameters from memory to GPU (the dominant cost in LLM inference), with the appropriate engineering effort, we can synthesize these weights on-chip from ~1000 geometric parameters! This trades compute for memory bandwidth—an extremely favorable exchange given that memory, not compute, is the primary bottleneck, as noted in the LIMINAL work (and other industry reports).
>
> As an aside, this reminds us of the Fast Fourier Transform: recognizing that the DFT matrix can be factored didn't immediately yield compute benefits, but this compositional structure later enabled $O(n \log n)$ algorithms compared to $O(n^2)$. Similarly, our decomposition reveals that dense linear layers have far fewer degrees of freedom than their raw parameter counts suggest.
>
> To summarize, we match expressivity with exponentially fewer parameters, which coincidentally achieves modest FLOP reduction, and more importantly, opens a path to eliminate memory bandwidth constraints (on the practical side) and to exploit the compositional structure (on the model development side).
>
>
> **Statement about limitations: The authors do not acknowledge limitations or potential negative societal impacts of their work.**
>
> We discuss limitations in Appendix A, but have now added a reference to this section in the main paper’s conclusion. Thanks for the suggestion.

---

> > ### Author Response · Authors · 2025-08-07
> > **Follow-up Comment**
> >
> > Thank you again for your thoughtful questions and feedback. As the discussion period is coming to a close, please let us know if there are any remaining questions--we'd be happy to clarify anything further.

---

> > > ### Comment · Reviewer_tqhq · 2025-08-09
> > >
> > > Thank you for your detailed response. I think it should be clarified that the proposed approach does not actually capture arbitrary linear maps. I remain positive about the paper.

---

> > > > ### Author Response · Authors · 2025-08-09
> > > >
> > > > Many thanks for you support of our work. Yes, we will make this clear for the reader.

---

### Note · Authors · 2025-08-11

We thank all reviewers for the constructive questions and discussion. We will incorporate all the proposed clarifications, end-to-end FMNIST experiments, analysis of computational complexity/cost, and additional discussion of future directions to further improve the paper.

---

### Decision · Program_Chairs · 2025-09-17

**Decision:**

Accept (poster)

**Comment:**

The authors present a method for parameterizing linear layers in neural networks in terms of compositions of geometric objects from Clifford algebra known as bivectors, and an algorithmic approach for decomposing these into simple factors in a differentiable way. The approach leads to an effective reduction in the number of parameters relative to the $d \times d$ baseline of a dense matrix, and empirical results in which a rotor-based decomposition is done to approximate the K, Q, and V dense matrices in a subset of LLM attention layers demonstrates comparable performance (in accuracy and perplexity) to strong baselines such as block-Hadamard and low-rank approximations, across different datasets. Initial reviews of the paper were positive -- authors were appreciative of the clear presentation, the convincing experimental results, and the potential for the technique to improve interpretability -- and increased further after clarifications and additional results during the rebuttal and discussion phase. I concur with the reviewers and recommend acceptance. The authors should carefully incorporate feedback from the reviewers during the rebuttal into the final revision, especially the promised additional experiment.